# Adaptive Scalable Video Streaming (ASViS): An Advanced ABR Transmission Protocol for Optimal Video Quality

Eliecer Peña-Ancavil *,†, Claudio Estevez †, Andrés Sanhueza † and Marcos Orchard †

Department of Electrical Engineering, Faculty of Physical and Mathematical Sciences, Universidad de Chile, Beauchef 850, Santiago 8370456, Chile; cestevez@uchile.cl (C.E.); asanhueza@ing.uchile.cl (A.S.); morchard@u.uchile.cl (M.O.)
* Correspondence: eliecer.pena@ug.uchile.cl
† These authors contributed equally to this work.

**Abstract:** Multimedia video streaming, identified as the dominant internet data consumption service, brings forth challenges in consistently delivering optimal video quality. Dynamic Adaptive Streaming over HTTP (DASH), while prevalent, often encounters buffering problems, causing video pauses due to empty video buffers. This study introduces the Adaptive Scalable Video Streaming (ASViS) protocol as a solution. ASViS incorporates scalable video coding, a flow-controlled User Datagram Protocol (UDP), and deadline-based criteria. A model is developed to predict the behavior of ASViS across varying network conditions. Additionally, the effects of diverse parameters on ASViS performance are evaluated. ASViS adjusts data flow similarly to the Transmission Control Protocol (TCP), based on bandwidth availability. Data are designed to be discarded by ASViS according to video frame deadlines, preventing outdated information transmission. Compliance with RFC 8085 ensures the internet is not overwhelmed. With its scalability feature, ASViS achieves the highest possible image quality per frame, aligning with Scalable Video Coding (SVC) and the available data layers. The introduction of ASViS offers a promising approach to address the challenges faced by DASH, potentially providing more consistent and higher-quality video streaming.

**Keywords:** SVC; streaming; VMAF; ASViS; ABR; flow control

## 1. Introduction

Over recent years, a marked transformation in the realm of digital communication has been observed, with multimedia video streaming emerging as the predominant internet data service. Some had predicted that by 2020, video traffic would account for 82% of global online traffic [1]. Contrary to these predictions, it is reported in [2] that in the first half of 2022, video streaming constituted 65.93% of total internet traffic, with Netflix and YouTube contributing 13.74% and 10.51%, respectively. Furthermore, a 23% increase in total traffic volume was observed in 2022 compared to 2021, attributed to the significant growth of various streaming services. Current data regarding video streaming consumption on mobile networks indicate that 71% of the traffic was video-related [3]. Projections suggest that this figure is expected to reach 80% by 2028. Such statistics underscore the undeniable growth trend. This evolution has been attributed not only to technological innovations but also to the integration of intelligent devices. Platforms, such as Zoom, Microsoft Teams, Netflix, YouTube, and Amazon Prime, have been recognized as significant players in this revolution. Moreover, unexpected external events, notably the Coronavirus disease pandemic, have acted as catalysts, propelling an unprecedented demand for online communication and entertainment platforms. This surge, documented as a 20–40% increase in video streaming [4,5], occasionally strained available bandwidth capacities, necessitating temporary transitions from high definition (HD) to standard definition (SD) [6] by some service providers.

Dynamic Adaptive Streaming over HTTP (DASH) is an open standard for transmitting video that adapts to both the network and device. In this evolving landscape, the DASH protocol, operating over the Transmission Control Protocol (TCP), has been identified as the primary delivery model for video streaming services [7–9]. This open-source standard, celebrated for its flexibility and codec-agnostic features, has established itself as an indispensable tool for most streaming entities [10,11]. Nevertheless, challenges associated with DASH, including buffering interruptions, have signaled an imperative need for refined protocols that can mitigate such disruptions [8].

In response to such challenges, focus within the research community has shifted towards optimizing Adaptive Bitrate (ABR) algorithms. One strategy involves having the client store chunks sent from the server on the local disk [7–9]. This stored content starts to play before the entire content is downloaded. It also helps tolerate network outages without interrupting playback by using the stored content on the local disk [12]. For clarity, the video content stored before playback on the client side is termed a Video Buffer (VB). This VB definition distinguishes it from the buffer term used in the context of the User Datagram Protocol (UDP) and Transmission Control Protocol (TCP) transmissions. The overarching objective remains clear: to develop a protocol that addresses DASH's inherent limitations and sets the groundwork for future streaming innovations. At the forefront of these explorative endeavors is Scalable Video Coding (SVC), a technique that, although previously underutilized, has been underscored for its potential to reshape the video streaming paradigm [13].

This paper introduces the Adaptive Scalable Video Streaming (ASViS) protocol, envisioned as a groundbreaking ABR transmission approach. At its core, ASViS leverages Scalable Video Coding (SVC) to partition video data into base and enhancement layers, with selective transmission based on layer-discarding policies. This differentiation is orchestrated by the application of a deadline-sensitive criterion. Specifically, each layer possesses a distinct "curfew", determining its transmission or omission based on the nearing of its playback deadline. For instance, as deadlines approach, higher-layer data might be excluded, ensuring the arrival of the base layers, which are more crucial for video coherence, are transmitted. ASViS, with its innovative feature set, promises not only to streamline video data transmission but also to elevate the end-user experience. The efficacy of ASViS is substantiated through comprehensive performance assessments, juxtaposed against the established Model Predictive Control (MPC) algorithm. Evaluation metrics, such as the Peak-Signal-to-Noise Ratio (PSNR) and the novel Video Multi-method Assessment Fusion (VMAF) technique by Netflix, are employed to gauge its performance.

For clarity and coherence, this paper is meticulously structured. Following this introduction, Section 2 delves into the intricacies of video coding and key quality metrics. Section 3 details the related work with the mechanics of streaming and ABR strategies. Section 4 elucidates the attributes and operational dynamics of ASViS, emphasizing its interplay with SVC. In Section 5, the simulation parameters adopted for ASViS evaluations are detailed, and the primary findings are presented. The paper concludes by drawing insights from the results, underscoring their significance in the evolving landscape of multimedia video streaming.

## 2. Background

### 2.1. Video Coding

The H.264 standard [14], also known as Advanced Video Coding (AVC) Part 10, is a predominant video compression standard in the industry [15,16]. This standard facilitates lossy compression by capitalizing on the similarity between adjacent pixels and the movement of objects across consecutive frames. Within AVC, a dependent frame structure termed a Group Of Pictures (GOP) is utilized [17]. Three types of frames are encompassed in AVC: Intra (I) frames, Predicted (P) frames, and Bidirectional (B) frames [8,18,19].

- **I frames**:
  - *Nature*: these frames encapsulate information solely from a single picture.
  - *Compression Method*: spatial redundancy within the picture is utilized for compression.
  - *Size and Independence*: typically larger in size, they can be decoded without relying on other frames.
  - *Significance*: aerve as reference points, with subsequent frame types drawing on information from I frames during the decoding process.
- **P frames**:
  - *Nature*: information from previous frames is used to predict and represent the current frame.
  - *Compression Method*: both spatial redundancy and temporal prediction (based on previously decoded frames) are employed.
  - *Efficiency*: encoding of P frames is more bit-efficient than I frames.
- **B frames**:
  - *Nature*: these frames interpolate between preceding and subsequent frames.
  - *Compression Method*: they make use of both forward (from subsequent frames) and backward (from prior frames) temporal predictions.
  - *Efficiency*: B frames typically represent the most bit-efficient frame type in the bit-stream.

Each GOP commences with an I frame and is followed by a sequence of P and B frames in a hierarchical manner. The GOP length signifies the interval between two I frames, inclusive of the initial I frame. Notably, the Transmission Order (TO) of video frames differs from the Display Order (DO), the latter of which pertains to the sequence during video playback. Dependencies are determined by hierarchy, with lower hierarchical frames depending on information from higher hierarchical frames for reconstruction. Figure 1 illustrates two types of hierarchical dependencies for a GOP of eight frames.

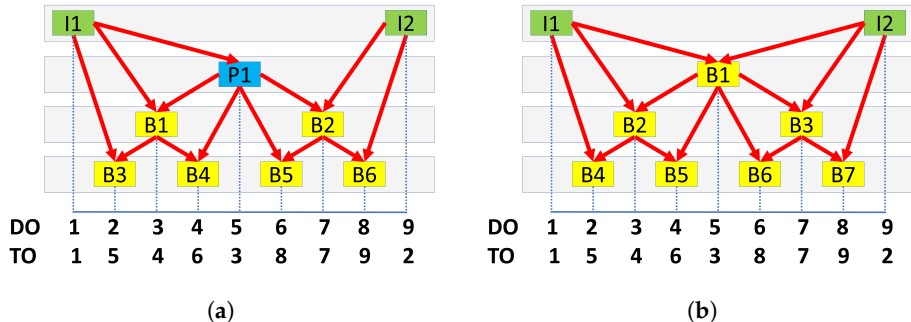

(a)            (b)

**Figure 1.** Hierarchy for (**a**) $I - P - B$ dependencies and (**b**) $I - B$ dependencies.

Progress in video coding techniques has introduced the concept of scalability. Within the AVC framework, this concept has been expanded with the introduction of SVC [16,20,21]. In SVC, frames consist of a Base Layer (BL) and one or more Enhancement Layers (EL). Each higher layer is contingent upon all the preceding layers for accurate display. If any layer is absent, subsequent layers are disregarded. SVC can be applied in terms of temporal resolution, spatial resolution, or quality. While SVC maintains the same GOP hierarchy as AVC, it introduces a parallel structure based on the number of ELs [22,23].

The term "scalability" in video streaming refers to a segmented layered coding system. This system allows for the removal of higher layers while still retaining the ability to decode the video [16,20]. Such a design minimizes information loss during packet drops, ensuring that videos maintain their integrity even with increased frame sizes.

SVC coding, in terms of the GOP hierarchy, mirrors the structure found in AVC. However, a distinguishing feature of SVC is its introduction of a parallel structure. This structure emerges from the number of ELs, all of which are contingent upon the BL [21–23].

A visualization of the SVC hierarchy for I–B with two layers and a GOP size of eight frames is provided in Figure 2, illustrating an IBBBBBBB pattern.

To determine the total number of layers, one must factor in the variety of frame types (I, B) and the number of SVC layers (such as BL, EL1, EL2, etc.). In the context of this research, four primary layers are discerned:

- Base I frame (BI)
- Enhancement I frame (EI)
- Base B frame (BB)
- Enhancement B frame (EB)

The base video data are housed within the BI and BB layers, while the enhancement video data reside in the EI and EB layers. For illustrative and experimental intents, this study adopts a GOP size of eight, following an IBBBBBBB (IB7) pattern, as depicted in Figure 1b. Two quality layers, BL and EL, are considered, with P frames being excluded from the analysis.

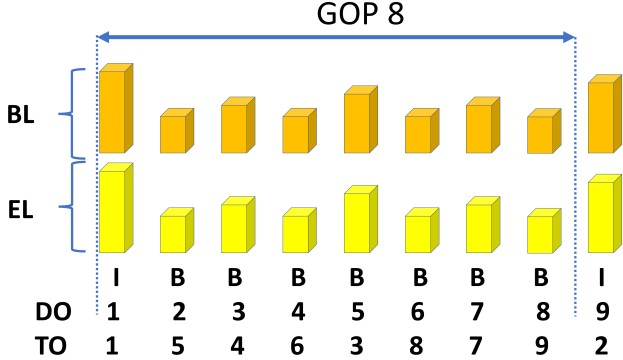

**Figure 2.** Hierarchy structure to a GOP of 8 for SVC with two layers.

Within the framework of SVC, the quality of a video frame is intrinsically linked to the layers employed during its reconstruction [20,23]. This relationship is exemplified in Figure 3, which provides a visual exploration of the impact on spatial Y-PSNR when specific frame types are omitted.

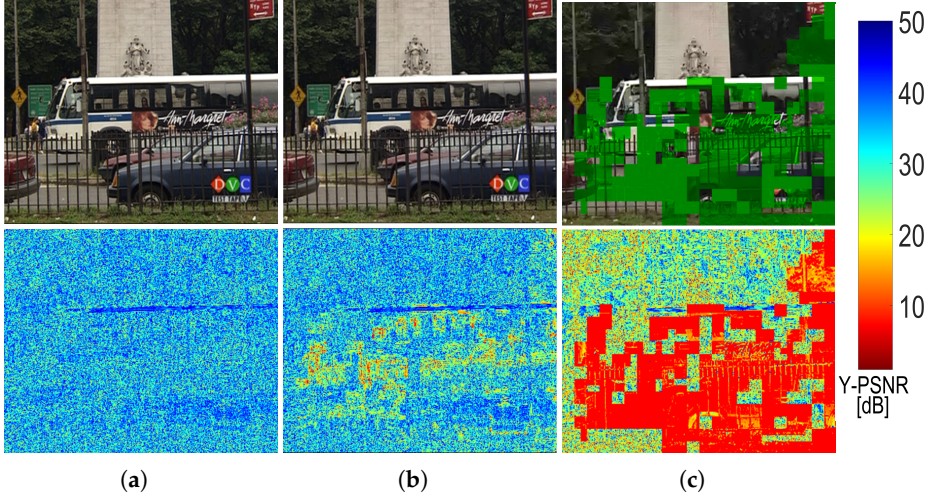

**Figure 3.** Effects of the loss of layers in the video quality for (**a**) a normal video, (**b**) no EL-B, and (**c**) Only BL-I.

Three distinct scenarios are considered in this test:

- **No Loss**: This scenario serves as a reference, wherein Figure 3a displays the spatial Y-PSNR solely resulting from frame compression and subsequent restoration.
- **Discarding All EB**: In this scenario, evident in Figure 3b, the frame exhibits regions with diminished Y-PSNR. This decrease is predominantly observed in areas undergoing pronounced color transitions, typically indicative of movement.
- **Retaining Only BI**: As portrayed in Figure 3c, this scenario experiences a significant information deficit. When both the BL and EL of a frame are absent, the resultant reconstruction contains voids. While it is feasible to populate these gaps with data from preceding frames, the primary aim of this experiment is to assess the impact without invoking supplementary measures.

### 2.2. Video Quality Measure

Determining video quality has long been debated due to its reliance on numerous subjective factors. The challenge lies in transforming these subjective perceptions into objective metrics [24,25]. Taking into account the human eye process of translating visible light into visual data, the prominence of photoreceptors becomes apparent [26].

Photoreceptors, crucial components of the human retina, facilitate the transformation of visible light into biological signals, allowing visual representation of the external environment [27]. The majority of these photoreceptors are rod cells, making up approximately 95% and predominantly recognizing luminance. On the other hand, cone cells account for approximately 5% and are chiefly responsible for color recognition [26,27]. Given this distribution, the human retina heightened sensitivity to brightness over color has led to video quality measures often concentrating solely on the luma channel.

### 2.2.1. Peak-Signal-to-Noise Ratio

The PSNR, a standard measure of video quality, characterizes the relationship between a signal maximal energy and its associated noise, presented in decibels (dB) [25,28]. Although both luma and chrominance components of a frame can be evaluated by PSNR, the emphasis is predominantly placed on the luma channel. The initial step in determining PSNR involves calculating the Mean Squared Error ($MSE$), expressed in Equation (1) [29]:

$$MSE = \frac{1}{mn} \sum_{i=0}^{m-1} \sum_{j=0}^{n-1} [I(i,j) - H(i,j)]^2 \tag{1}$$

In this equation, pixel values of the original and reconstructed frames are represented by $I(i,j)$ and $H(i,j)$, respectively, with $m$ and $n$ indicating the frame horizontal and vertical dimensions. PSNR can then be calculated using Equation (2), with $MAX_I$ representing the frame maximum pixel value:

$$PSNR = 10log_{10}\left(\frac{MAX_I^2}{MSE}\right) \tag{2}$$

While PSNR has been a foundational metric in video quality assessment, its capability to fully represent the subjective perception of end users has been a topic of scrutiny. This is due to its focus on mathematical differences at the pixel level, which does not always align with human perception. Especially in adaptive streaming systems, there is a pressing need for metrics that can more accurately reflect human experiences.

### 2.2.2. VMAF

In response to these limitations, Netflix introduced VMAF, a video quality metric crafted to mirror human perception more accurately [25,30]. By amalgamating various video quality evaluation methods using a Support Vector Machine (SVM), VMAF produces a perceptual quality score [12,24,31]. This integrated approach ensures the holistic representation of individual measures, bolstering its correlation with subjective assessments.

Validations across a multitude of video resolutions and datasets indicate VMAF scores, ranging between 0 and 100, align closely with subjective evaluations [31,32].

To compute VMAF, spatial and temporal video quality measures are initially extracted as feature maps [12,31]. Spatial metrics are derived from the Detail Loss Metric (DLM) and Visual Information Fidelity (VIF), while the temporal metric employs Temporal Information (TI). Detail losses, representing the absence of crucial visual data impacting content clarity, are evaluated across four scales using DLM. In contrast, VIF discerns these losses and visual information fidelity across four distinct scales. TI gauges the luminance difference between frame pairs to identify temporal variations resulting from movement, encapsulated by six unique features [31]. Following this, the average value of each feature is ascertained and channeled into a pre-trained SVM, which then predicts the per-frame quality score, with 0 denoting low quality and 100 signifying the highest quality [32]. The VMAF underlying model was developed using SVM regression on the VMAF+ video database. VMAF+ is a comprehensive Video Quality Assessment (VQA) dataset, comprising 522 videos subjected to diverse scaling and compression levels [33]. During the supervised learning phase, the VMAF model ascertains optimal weights for rudimentary metrics, learning from established ground truth scores [32]. These steps are portrayed in Figure 4. Consistent with other advanced video quality metrics, VMAF predominantly assesses the luma components of the video channel [33].

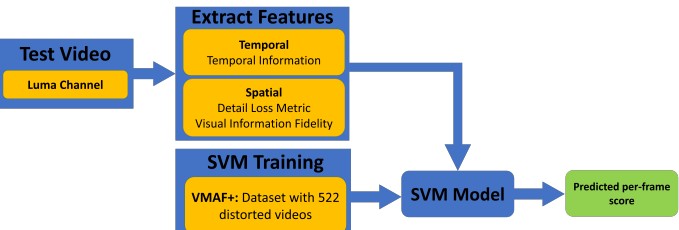

**Figure 4.** Outline of VMAF algorithm.

In adaptive streaming systems such as DASH, accurate video quality metrics become pivotal. Such systems, in real time, determine the optimal video quality to transmit, considering network conditions and device capabilities. A metric misaligned with user perception might lead to suboptimal streaming decisions, either compromising user experience or wasting bandwidth. Thus, the incorporation of user-centric metrics is imperative for effective adaptive streaming decisions.

## 3. Related Work

### 3.1. Dash

DASH is known as an open-source standard, offering a modular approach for the delivery of video content between the client and the server. By using Hypertext Transfer Protocol (HTTP) for data transmission, firewall-associated challenges are mitigated, data delivery is assured, and compatibility with a plethora of network protocols is fostered [34,35].

On the server end, videos are typically segmented into K chunks with intervals usually ranging between 1 and 5 s [36,37]. Each chunk can be represented by multiple GOPs using conventional I–P–B or I–B frame hierarchies. Each chunk is encoded based on various quality parameters such as quantization, frame rate, and resolution. It is noted that encoding at heightened resolutions or quantizations requires increased bitrates.

From the client perspective, the ABR algorithm plays a crucial role, dynamically determining the bitrates for each chunk during playback. Once a chunk is initiated for download, transitions between chunks are not possible until the current chunk finishes. The transport layer, susceptible to variations due to fluctuating network conditions, ensures the receive buffer is populated at the rate of available throughput [38]. Achieving balance between the receive and extract rates is vital for a continuous video experience. Nonetheless, network-induced variations can lead to:

- A lag in enhancing video quality following improved network conditions, thereby underutilizing the available throughput.
- Potential interruptions in playout if a chunk of higher bitrate is downloaded amidst unfavorable network conditions.

A pivotal consideration in DASH remains the playout frame deadline, hereafter referred to as the deadline. This deadline pinpoints the time by which video frames should be decoded, ensuring a seamless viewing experience. Within a DASH framework, the transmission interval between the server and the client is integral.

Central to DASH, the ABR algorithm aspires to optimize the resolution per frame, facilitate smooth video playout, and ensure timely frame delivery [39]. However, achieving these objectives concurrently can be intricate given the constraints of bandwidth and ever-changing network conditions.

ABR Algorithm Approaches

Various classifications exist for ABR algorithms, including throughput-based, buffer-based, and a synthesis of the two, commonly termed the hybrid approach [40]. It is observed that a majority of ABR algorithms derive their decision-making from a balance between network conditions and video coding nuances. Throughput-focused strategies often use the bandwidth data of the application layer to navigate the choice of bitrate chunks. Techniques such as last-packet download and sliding window moving average are often employed to estimate available bandwidth [41]. However, these approaches may grapple with issues of transient fluctuations in bandwidth.

On the other hand, buffer-centric strategies adopt thresholds within the VB occupancy to make decisions [41]. These techniques underscore the significance of absorbing throughput variations and retaining sufficient content in the VB to avert interruptions [42]. Hybrid strategies typically select bitrate chunks centered around a Quality of Experience (QoE) prediction metric [12,40]. Such metrics might encompass parameters such as throughput, VB status, and even details related to employed technology.

*3.2. Modern Video Streaming Solutions*

Contemporary research has proposed a plethora of techniques to elevate video streaming performance [43,44]. Solutions, such as the one in [43], implement a finite state machine at the application layer. This strategy deploys bandwidth estimation, VB size on the client side, and individual SVC layer encoding rates to discern the layered blocks for transmission.

Contrastingly, techniques detailed in [44] harness multiple paths to transmit distinct blocks and layers, enhancing the probability of base layer reception. [45] introduces a decision support system, emphasizing the minimization of quality fluctuations between adjacent chunks, potentially elevating user QoE.

Other notable solutions, such as [13], capitalize on software-defined networks to enhance DASH adaptability. While a myriad of physical-application cross-layer solutions such as [46] exist, transport-application cross-layer solutions remain relatively scarce.

In [47], a collaboration between physical, Media Access Control (MAC), and application layers is observed. This alliance aligns the performance of the lower layer data rate with the application layer bitrate via machine learning. Moreover, [48] provides a discourse on the fairness challenge in TCP video transmissions.

Model Predictive Control Algorithm in Video Streaming

MPC, introduced in [49], emerges as a state-of-the-art technique for optimizing QoE in video streaming. MPC presents a hybrid approach, formulating the video ABR algorithm as a stochastic optimal control problem. By leveraging both throughput and VB strategies, it seeks an optimal balance [50]. This methodology employs a sliding look-ahead window to forecast a series of pivotal parameters. The derived predictions are subsequently employed to tackle the optimization problem, aiming to enhance QoE [50].

Several key parameters, pivotal for QoE, have been identified. These parameters encompass the size of chunk $k$ encoded at bitrate level $R$, denoted as $d_k(R_k)$, perceived video quality ($q$), throughput ($C$), and the VB occupancy at the client end ($B$) [51].

QoE is articulated using a spectrum of metrics. Among these, the Average Video Quality ($AVQ$) stands as a prime metric, reflecting the overall perceived quality over all chunks:

$$AVQ = \frac{1}{K} \sum_{k=1}^{K} q(R_k) \tag{3}$$

The subsequent metric, average video quality variation ($AVQV$), quantifies the shifts in video quality between sequential chunks:

$$AVQV = \frac{1}{K-1} \sum_{k=1}^{K-1} q(R_{k+1}) - q(R_k) \tag{4}$$

Rebuffering ($Re$) stands as a crucial metric, expressing the interplay between the download time of a chunk and the occupancy of the client VB:

$$Re = \sum_{k=1}^{K} \left( \frac{d_k(R_k)}{C_k} - B_k \right)_+ \tag{5}$$

It should be noted that rebuffering occurs when the download time of a chunk exceeds the VB occupancy level. The startup delay, represented by $T_s$, signifies the initial waiting period before playback—a phase often utilized to preload the VB and assess the connection quality. The holistic QoE, spanning chunk 1 through $K$, is synthesized as:

$$QoE_1^k = AVQ - \lambda \cdot AVQV(K-1) - \beta \cdot Re - \beta_s \cdot T_s \tag{6}$$

Here, weight parameters $\lambda$, $\beta$, and $\beta_s$ can be tailored based on the relevance of each QoE component. The notation $(x)_+ = max\{x, 0\}$ ensures that the rebuffer parameter in Equation (5) remains non-negative.

## 4. Proposed Cross-Layer Solution: ASViS

### 4.1. Overview of ASViS

ASViS, or adaptive scalable video streaming, is a cross-layer solution that bridges the gap between the transport and application layers. Within the application layer, an algorithm is implemented that manages the dispatched information, relying on a layer-discarding policy and deadline-sensitive criteria. While packet loss assists in determining the flow rate, reflecting the behavior of TCP, retransmissions are not essential. Based on the layer-discarding policy, certain packets might be omitted. Utilizing UDP, the system detects missing frame fragments at the application layer. Since there is no requirement to track sequence numbers or acknowledge every packet, UDP becomes a preferred choice. The flow rate is managed by adjusting the number of packets dispatched over a time span. This approach, similar to the congestion window of TCP, is termed here as the flow window or fwnd. ASViS supports Selective Acknowledgment (SACK); however, it is dispatched only when deemed essential. SACK packets are vital for measuring round-trip time ($RTT$), which is required for estimating available throughput. Lost packets might be retransmitted if the time and priority permit. The flow window is principally employed to fulfill with RFC 8085 requirements, which emphasize preventing network saturation [52].

Scalability is an important factor in the growth of any technology. When referring to ASViS, scalability is characterized by several factors: 1. Since ASViS is based on SVC, which itself is part of H.264 (and newer), it can easily evolve to higher resolutions (8 K, 16 K, etc.). 2. SVC does not have a fixed amount of layers; to prove our concept, this work uses four SVC layers, but more can be used. As technology progresses and resolutions increase, it is beneficial to also increase the amount of SVC layers used, since it would create more

granularity (i.e., more quality options to choose from). There are other aspects not covered in this work, in which ASViS can scale (such as GOP size, block compression size, etc.) that are inherent to the compression standards. All these degrees of scalability allow ASViS to maintain itself relevant in the future.

Compared to traditional video streaming methods based on TCP, which often rely heavily on retransmissions and sequence acknowledgments, ASViS, with its refined layer-discarding policy and strategic use of UDP, offers a more efficient and adaptable approach. This methodology not only reduces overhead but also aligns closely with the RFC 8085 guidelines. These guidelines are a series of recommendations on UDP usage and best practices designed to prevent network congestion. By complying with these guidelines, stability and efficiency in network operations are ensured. The adherence of ASViS to this standard is underscored by its significance.

A deadline-sensitive approach is employed, which operates on a time-based metric. This metric is focused on the residual time to meet the playout frame deadline. In SVC codification, by default, all layers of a frame share the same deadline. This approach introduces differentiation, suggesting the omission of certain layers based on pre-set criteria using the deadline as a reference. A significant challenge in video streaming emerges when throughput is below the video bitrate. Stream prioritization becomes essential, with low-priority data being discarded as deemed necessary. The inherent structure of SVC layers, where higher layers rely on foundational ones, designates a higher priority to base layers. As the deadline becomes imminent, high-layer data are omitted, and if the situation demands, additional layers are also removed.

The proposed layer-discarding policy suggests having differentiated deadlines for distinct layers. To distinguish it from the frame deadline, the layer deadline is referred to as *curfew* . Higher layers (lower priority) have earlier curfews, and lower layers (higher priority) have more lenient curfews. The curfew spacing is an important design parameter. The algorithm may accept a low-priority layer at the cost of running out of time to process a subsequent higher-priority layer if a late curfew (short present-to-deadline period) is used. However, if the curfew is too early, the algorithm response may be premature, and higher-layer (enhancement) packets can be unnecessarily discarded. The decision-making process is a critical part of the proposed solution. It differentiates discarding criteria to specific layers for an SVC video with two layers (BI, BB, EI, and EB), as shown in Figure 5 where all layers of a frame have a defined threshold.

- Data received after the deadline are discarded, regardless of their priority.
- Only BI layers are considered for data that arrive between the deadline and the first curfew.
- At the second threshold, both BI and BB layers are accepted.
- When the third threshold is reached, BI, BB, and EI layers are permitted.
- Beyond the third threshold, all layers, inclusive of EB, are accepted.

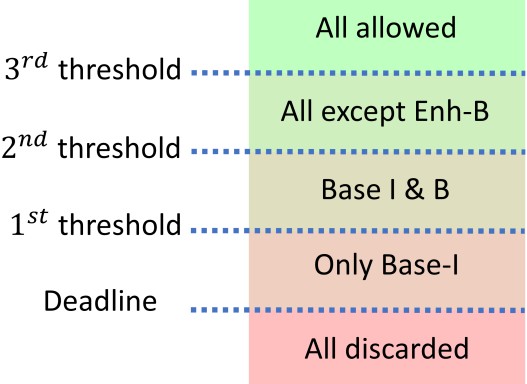

**Figure 5.** Thresholds for an SCV video with 2 layers.

Deadline-Sensitive Criteria Algorithm (DSCA)

The decision-making process, shown in Algorithm 1, consists of reading a packet and extracting the frame (I or B) and layer (base or enhancement) information. Based on this, it computes the curfew, which considers the current time, frame deadline, travel time, and priority. The curfew is computed only on packets that are inside the current *fwnd* , as later packets depend on current decisions. Higher-priority packets have curfews closer to the deadline, while lower-priority packets are discarded earlier to increase the chances of a higher-priority packet arriving on time. If a packet inside the current *fwnd* is discarded, a later packet is pushed to an earlier time, increasing its probability of successfully arriving before the deadline. This is the fundamental principle, increasing the chances of successfully arriving before the deadline.

---

**Algorithm 1** Packet Decision Process

---

```
 1:  while process not ended do
 2:      ObtainNextPacketInfo(frame, layer)
 3:      if ArrivesBeforeDeadline(frame, layer) then
 4:          if IsLastPacket() then
 5:              EndProcess()
 6:          else
 7:              SendPacket(frame, layer)
 8:          end if
 9:      else
10:          DiscardPacket(frame, layer)
11:      end if
12:  end while
```

---

*4.2. Modeling ASViS*

The quality of received video is contingent on available bandwidth. ASViS is designed to manage packet discards when bandwidth is limited. At the application layer, flow control is implemented, similar to TCP behavior. However, unlike TCP, ASViS allows the sender to decide on packet discards, which is why UDP is utilized. The receiver responds with SACK, enabling the sender to adjust the flow upon detecting packet losses.

The initial step involves mapping video data to packets. The simulation assumes that each video frame is transmitted in an individually assigned packet, or multiple packets if the frame size exceeds the Maximum Segment Size (*MSS*). However, multiple video frames are not encapsulated in a single packet to avoid losing multiple frames with a single packet loss. To maintain clarity, we use the term "frame" strictly in the context of a video image, distinguishing it from a data-link layer unit.

The video bitstream is denoted as $b_S = \frac{d_T}{t_T}$, where $d_T$ represents the total data to be transmitted, including all layers, and $t_T$ is the total encoding time. Total data can be expressed as:

$$d_T = \sum_{f=1}^{F_T} \sum_{l=1}^{L} d_{\langle f,l \rangle}. \tag{7}$$

$F_T$ is the total amount of frames and $L$ is the total amount of layers per frame. $d_{\langle f,l \rangle}$ is the amount of data needed to transmit the $l^{th}$ layer of the $f^{th}$ frame. The bitstream is varied according to the available bandwidth, considering discarded packets, so the effective bitstream is:

$$b_F = \frac{d_F}{t_T} \text{ and } d_F = \sum_{f=1}^{F_T} \sum_{l=1}^{l_F} d_{\langle f,l \rangle}. \tag{8}$$

$l_F$ is the number of layers transmitted for frame $f$, where $l_F$ could have a value of 0 if the whole frame is discarded. Since the transport layer is flow-controlled, it is necessary to determine the number of packets required. The flow can be controlled on a packet or byte basis. In most cases, protocols operate on a packet basis. The amount of packets necessary to transmit all the frames that are not discarded (arrive successfully) is:

$$S_{\langle f,l \rangle} = \left\lceil \frac{d_{\langle f,l \rangle}}{MSS} \right\rceil. \tag{9}$$

To simplify the frame and layer indexing to a single indexing variable, the following conversion can be made: $q = L(f-1) + l$, where $L$ is the total amount of layers, $l$ is the layer index per frame spanning from 1 to $L$, and $f$ is the frame index varying from 1 to $F_T$. Therefore, $q$ spans from 1 to $L F_T$, and (3) becomes $S_q = \lceil d_q / MSS \rceil$. The *fwnd* varies with each *RTT* cycle depending on whether packets arrive successfully at the client side. The *fwnd* is represented by a vector where each entry corresponds to an *RTT* cycle $w$, where $W$ is the total number of cycles, i.e., $\overrightarrow{fwnd} = \{fwnd_1, \dots, fwnd_w, \dots, fwnd_W\}; w \in \mathbb{N}$.

Given the packet $q$, the function $C_q$ outputs the *RTT* cycle $w$ where the last packet of layer $q$ is found. For better accuracy, a packet-acceptance pattern should be included, which is a set of ones and zeros representing true and false statements, respectfully, responding to whether the packet successfully arrived at the destination and before the deadline. This is represented by the vector $\overrightarrow{X} \in \{0,1\}$. There is a correlation between the number of ones in $\overrightarrow{X}$ and the overall quality of the video.

To model the behavior of the packet-acceptance pattern, it is necessary to perform an iterative computation to determine if that layer is to be discarded or retained.

The **first step** involves determining the deadline and curfew of the layer. To determine this, the *RTT* cycle is needed, and it is given by $C_q$:

$$C_q = \arg\min_w \left\{ \sum_{n=1}^{w} fwnd_n - \sum_{\hat{q}=1}^{q} S_{\hat{q}} X_{\hat{q}} > 0 \right\} \tag{10}$$

$S_q$ is the number of packets required to send layer $q$. $X_q$ is the $q^{th}$ value of the packet-acceptance pattern vector, but for this stage, we assume $X_q = 1$. All values earlier than $X_{q-1}$ (inclusive) are known.

**Second step**. With the *RTT* cycle $C_q$ information, the estimated arrival time of the packet $q$ (or $t_{A_q}$) is computed as:

$$t_{A_q} = RTT \cdot (C_q|_{X_q=1} - 1/2) \tag{11}$$

**Third step**. The deadline of frame $f$ (or $D_f$) is a straight-forward computation obtained using the buffer time $t_B$ and the frame rate $r$, hence $D_f = \frac{f}{r} + t_B$. It is beneficial to obtain a deadline expression in terms of the packet index $q$ (or $D_q$). Since all layers of a frame have the same deadline, the relation $f = \lfloor q/L \rfloor$ is sufficient to compute $D_q$. Substituting gives:

$$D_q = \left\lfloor \frac{q}{L} \right\rfloor \frac{1}{r} + t_B \tag{12}$$

**Fourth step**. To retrieve the priority of packet $q$ (or $p_q$) for $P$ levels of priority, it is necessary to know the GOP pattern. As mentioned earlier and shown in Figure 1, the GOP pattern is IBBBBBBB. To obtain the priority level of this pattern as a function of layer $q$, the following expression is used:

$$p_q = \begin{cases} 2(q-1) & \text{if } q \leq L \\ 2\,\Pi(L) & \text{if } q > L \,\&\, \Pi(L\,G) < L, \\ 2\,\Pi(L)+1 & \text{if } q > L \,\&\, \Pi(L\,G) \geq L \end{cases} \tag{13}$$

where $\Pi(\lambda) = mod(q - L - 1, \lambda)$ and $G$ is the GOP size, in this case, $G = 8$. $p_q$ spans from 0 to $P - 1$, where the lower values have higher priorities. Note: $q$ is in the coding order (see Figure 1).

**Fifth step**. Once the priority is found, the curfew $\Gamma_q$ is determined on Equation (14), where the number of priority levels is obtained by $P = L\,T$ where $T$ is the number of frame types. Here, we work with I and B frames, so $T = 2$.

$$\Gamma_q = mod(p_q, P) \cdot \tau \tag{14}$$

$\tau$ is a positive value in units of $RTT$ that determines the time spacing between priority levels, and it is a design parameter. The greater the value, the more conservative the algorithm, and the lower is riskier. Increasing $\tau$ increases the chances of having a smooth video stream, lowering $\tau$ attempts to transmit a higher quality (more layers) stream but with a higher probability of loss due to limited bandwidth.

**Final iteration step**. The Packet-Acceptance Pattern (PAP) set can be obtained as the transmission conditions are met using:

$$X_q = \begin{cases} 1 & \text{if } D_q - t_{A_q} - \Gamma_q \geq 0 \\ 0 & \text{if otherwise} \end{cases}. \tag{15}$$

The value of $X_q$, which at the beginning of the iteration was assumed to be 1, is now determined. This value is used for the subsequent iteration.

Optimization Method for $\tau$ Configuration

In the study presented in [53], a proof-of-concept is demonstrated in which video quality is maintained without re-buffering. However, an in-depth formal analysis for optimizing the parameter $\tau$, representing curfew gaps, is not provided. An exploration into varying $\tau$ values within a reasonable range reveals that the maximum Y-PSNR is indicative of optimal video quality.

For optimization scenarios aiming at a single objective such as a root, maximum, or minimum, the bisection method is commonly employed [54,55]. However, its limitation in potentially identifying local maxima instead of the global maximum is acknowledged. Therefore, a multi-section variant of the bisection method is utilized in this study.

A Multi-dimensional Multi-section (MM) method is introduced, extending the bisection method for multi-dimensional optimization while minimizing the likelihood of identifying local maxima. Unlike traditional bisection methods, which divide the search space into two [54,55], a more expansive search technique is adopted here to mitigate the risk of mistaking local maxima for global ones.

Various applications show a monotonic increase up to a maximum, followed by a monotonic decrease, making traditional methods effective. However, for systems exhibiting oscillatory magnitudes, a more cautious approach is warranted. In this study, multiple points per dimension are probed in each iteration, with the step size and search region being adjusted based on the identified maximum. Achieving satisfactory result precision within a reasonable computational time is a non-trivial task, particularly due to the discrete nature of the data. The algorithmic details of the multi-probe bisection method are presented in Algorithm 2.

---

**Algorithm 2** Multi-dimensional Multi-section Method (3D example)

---

1: $[m_1, m_2, m_3] \leftarrow [0,0,0]$
2: $[M_1, M_2, M_3] \leftarrow$ upper limit of $\tau_1, \tau_2, \tau_3$
3: $range_x \leftarrow (M_x + m_x)/2$ **for** $x = 1, 2, 3$
4: $N \leftarrow$ *number of steps*
5: $step_x \leftarrow range_x/(N-1)$ **for** $x = 1, 2, 3$
6: $Q \leftarrow$ *empty matrix* $(1 \times 0)$
7: $list_{\tau_1, \tau_2, \tau_3} \leftarrow$ *empty matrix* $(3 \times 0)$
8: **while** error tolerance (init $i \leftarrow 0$ before loop)
9:   **for** $\tau_1$ **from** $m_1$ **to** $M_1$ **step** $step_1$
10:     **for** $\tau_2$ **from** $m_2$ **to** $M_2$ **step** $step_2$
11:       **for** $\tau_3$ **from** $m_3$ **to** $M_3$ **step** $step_3$
12:   **if** $[\tau_1, \tau_2, \tau_3]$ **is found in** $list_{\tau_1, \tau_2, \tau_3}$ **then** *skip*
13:   **else** (increase $i \leftarrow i + 1$)
14:     **add** $[\tau_1, \tau_2, \tau_3]$ **to** $list_{\tau_1, \tau_2, \tau_3}$ **in row** $i$
15:     **run** test of video quality
16:     **add** quality result **to** $Q$ **in row** $i$
17:   **end if**
18:     **end for**
19:     **end for**
20:   **end for**
21:   $range_x \leftarrow range_x/2$ **for** $x = 1, 2, 3$
22:   $step_x \leftarrow step_x/2$ **for** $x = 1, 2, 3$
23:   $Q_{MAX} \leftarrow$ max$(Q)$
24:   $i_{MAX} \leftarrow$ indices of $Q$ where $Q = Q_{MAX}$
25:   $list_{\tau_1, \tau_2, \tau_3}^{MAX}$ **extract rows** $i_{MAX}$ **from** $list_{\tau_1, \tau_2, \tau_3}$
26:   $m_x \leftarrow$ min$(\tau_x) - range_x$ **for** $x = 1, 2, 3$
27:     **where** $\tau_x$ **belongs to** $list_{\tau_1, \tau_2, \tau_3}^{MAX}$
28:   $M_x \leftarrow$ max$(\tau_x) + range_x$ **for** $x = 1, 2, 3$
29:     **where** $\tau_x$ **belongs to** $list_{\tau_1, \tau_2, \tau_3}^{MAX}$
30: **end while**

---

## 5. Experimental Setups and Results

Experiments were conducted in a structured manner to evaluate the performance of the ASViS algorithm in comparison to conventional methods. Each investigation was targeted towards elucidating specific attributes and behaviors. The performance of ASViS is influenced by several specific parameters, which will be elaborated upon in the subsequent sections. Notably, network conditions, such as *RTT*, packet loss, and buffer conditions, have been identified. Additionally, variations in $\tau$ parameters across different layers and alterations in the size of each layer have been observed to impact performance. A thorough examination of these factors and their implications on ASViS functionality is included in the following discussions. The first experiment juxtaposed a theoretical model of ASViS, particularly in assessing the anticipated arrival time of frames at the client side, against an empirical counterpart. The purpose was to determine the accuracy of the theoretical model in simulating network dynamics. The second experiment delved into the perceived video quality by modulating the parameter $\tau$. By setting a continuum of equidistant benchmarks, the Y-PSNR was gauged, revealing a complex interrelation between the $\tau$ values and the Y-PSNR attained. In the third analysis, the objective was to discern the optimal curfew gaps, represented as the set of taus, suitable for static initial network scenarios. Here, the MM method paved the way to unearthing superior video quality. Unlike assuming uniform curfew gaps, this study hypothesized better outcomes from distinct-sized intervals, with the Y-PSNR being the primary metric of evaluation. The fourth exploration sought to validate the theory that layer-based deadline gaps echo the proportions of the layer sizes. The culminating experiment instituted a face-off between two ABR protocols, ASViS and MPC. Using a spectrum of metrics, from VB behavior and throughput to video quality,

this comparative study decoded the operational nuances of ASViS juxtaposed with the MPC algorithm.

The video utilized for simulations comes from the Xiph video repository [56], specifically the bus 352 × 288 .yuv file, encoded at 30 fps. The Joint Scalable Video Model (JSVM) facilitated video coding and decoding on SVC. Video details and parameters are provided in Table 1. Insights into parameters of coded video files are provided by the JSVM trace file, which facilitates video reconstruction using specific layers or packets. The ASViS algorithm was implemented using MATLAB and JSVM. Different network conditions were simulated based on the layer information extracted from the JSVM trace file, determining whether packets were transmitted or discarded. Furthermore, MATLAB was employed in conjunction with JSVM for tasks associated with video encoding/decoding. Additionally, the results were organized and processed using MATLAB. JSVM works using the operating system command prompt. It is easy to link JSVM to MATLAB using the matlab *system* command.

**Table 1.** Video parameters for all experiments.

| Parameter | Details |
|---|---|
| File | bus_cif.y4m |
| Rate | 30 fps |
| Length | 150 frames |
| Number of layers | 2 |
| Quantization Layer 1—BL | 40 |
| Quantization Layer 2—EL | 20 |
| GOP | 8 |
| Resolution | width 352, height 288 |
| Amount of BI/EI | 19 |
| Amount of BBI/EB | 121 |
| Y-PSNR (Orig. Encoded) | 40.62 [dB] |
| VMAF (Orig. Encoded) | 99.97 [%] |

The video quality of different ABR algorithms was gauged using PSNR, and for VMAF testing, version 0.6.1 was employed from github (https://github.com/Netflix/vmaf), incorporating default pre-trained machine learning coefficients.

### 5.1. Results

The objective of the first experiment is to compare the network behavior outcomes of a theoretical ASViS model with those of an experimental test. Both scenarios employ UDP transmission rules equipped with flow control to prevent congestion. In the theoretical model, the determination of which packets are discarded is based on the Estimated Packet Size (EPS) and network conditions as utilized by the ASViS algorithm. The EPS values are informed by the mean layer size information derived from a trace file, which is created for JSVM. The size in *MSS* for each layer is presented in Table 2. For the experimental model, the actual size of each layer, influenced by its content, whether it is BL or EL, and its classification as I or B, is utilized. This results in a variety of sizes for each layer. The size in [kB] for each layer is presented in Table 2. Both models operate with the same network transmission parameters: *RTT* 0.05 s, packet loss rate 0.01, and buffer 3 s. Results are calculated based on 100 pseudo-random seeds.

**Table 2.** EPS for each layer for ASViS theoretical experiment.

| Layer | EPS [kB] |
|---|---|
| *BI* | 7154 |
| *BB* | 584 |
| *EI* | 36,500 |
| *EB* | 11,388 |

Within the ASViS model, the behavior can shift from conservative to risky by adjusting the $\tau$ values. For a video comprising four layers, it might be presumed that there would be four distinct $\tau$ values representing BI, BB, EI, and EB. However, in this experiment, the $\tau$ value for the *BI* layer is set at 0 due to its importance. This ensures its transmission until the deadline is reached. Seven configurations, each with an equidistant and incremental $\tau$ gap ($\tau$G) based on layer priorities, are outlined in Table 3.

**Table 3.** 7 $\tau$ gap configurations to ASViS.

| $\tau G$ | $\tau_2$—BB [RTT] | $\tau_3$—EI [RTT] | $\tau_4$—EB [RTT] |
|:---:|:---:|:---:|:---:|
| $\tau G_1$ | 1 | 2 | 3 |
| $\tau G_2$ | 2 | 4 | 6 |
| $\tau G_3$ | 3 | 6 | 9 |
| $\tau G_4$ | 4 | 8 | 12 |
| $\tau G_5$ | 5 | 10 | 15 |
| $\tau G_6$ | 6 | 12 | 18 |
| $\tau G_7$ | 7 | 14 | 21 |

The results, both theoretical and experimental, appear in Figure 6. This representation showcases the estimated arrival time of each frame at the client end. The gradient of the slope conveys the transmission rate; a gentler slope indicates a higher rate, whereas a steeper one suggests a slower rate. A significant gap between theoretical and experimental results is observed for $\tau G_7$. This discrepancy can be attributed to various factors, including differences in average vs. actual packet size and video complexity variations.

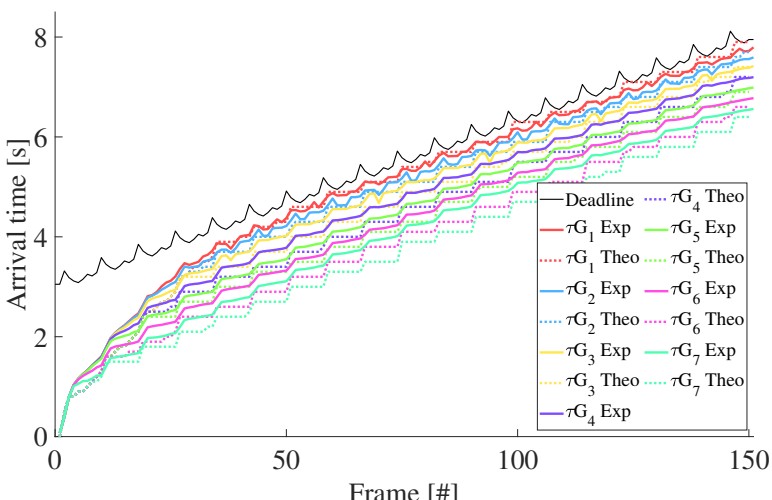

**Figure 6.** Results of the arrival time of frames for theoretical and experimental scenarios.

To determine the accuracy of the correlation between models, the Mean Absolute Percentage Error (MAPE) is employed. Table 4 provides a detailed breakdown, pointing out a maximum prediction error of approximately 14.2% for $\tau G_6$. Given the inherent variability of the practical experiments due to packet losses and the intricate characteristics of each frame, along with the nuances of the various layers and complexities associated with network conditions in empirical analyses, the accuracy of the theoretical model is underscored. It is noteworthy that high fidelity is achieved by the theoretical model, aligning well with the experimental results. This contributes to a reduction in the need for exhaustive and computationally demanding empirical tests, thereby facilitating more streamlined behavioral predictions based on the theoretical model. It is important to stress that this does not replace empirical benchmarks but can significantly reduce the amount of energy, resources, and time devoted to more practical tests.

**Table 4.** MAPE comparison from theoretical and experimental results of ASViS.

| $\tau G$ | MAPE [%] |
|----------|----------|
| $\tau G_1$ | 6.5 |
| $\tau G_2$ | 7.3 |
| $\tau G_3$ | 9.3 |
| $\tau G_4$ | 10.7 |
| $\tau G_5$ | 10.1 |
| $\tau G_6$ | 14.2 |
| $\tau G_7$ | 13.7 |

The aim of the second experiment is to analyze how the video quality performance of ASViS is influenced by varying $\tau$ values, placing emphasis on the experimental outcomes. The network transmission parameters for this investigation are as follows: *RTT* 0.05 s, packet loss rate 0.01, and buffer 1 s. Results are calculated based on 100 pseudo-random seeds for each $\tau G$. The configuration of $\tau G$ adheres to the format provided in Table 3. This study also assesses the behavior of SVC in the absence of ASViS, termed the "no protocol" condition.

Figure 7 delineates the arrival time at the client side against each frame for every specified $\tau G$. The scenario devoid of a protocol displays results closely aligned with the deadline. Nevertheless, the Y-PSNR performance for each $\tau G$ is illustrated in Figure 8. This representation indicates superior performance in the "no protocol" scenario. It is also observed that there exists a significant correlation between increased $\tau G$ and decreased Y-PSNR.

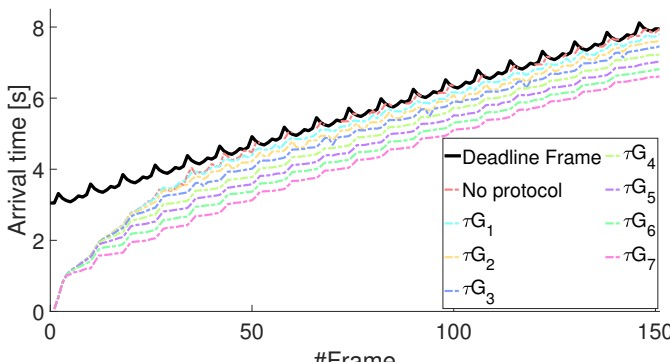

**Figure 7.** Results of the arrival time of frames for $\tau G$ and no protocol scenarios.

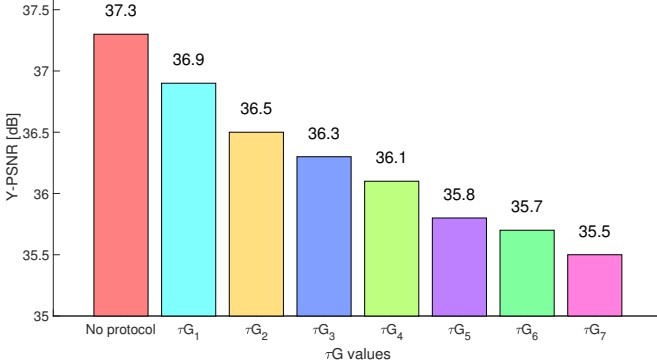

**Figure 8.** Y-PSNR performance for $\tau G$ and no protocol scenarios.

At an initial assessment, it may be surmised that SVC without ASViS offers superior results. However, upon closer analysis, this assumption is challenged. One primary objective of ABR algorithms is the prevention of rebuffering. Data presented in Figure 9, which portray the fluctuation in VB size relative to video playout, reveal stalls occurring

after the 50% playout in scenarios lacking a protocol. In situations with ASViS, there are no observable stalls for any $\tau G$, and a rise in the VB size is witnessed as the $\tau G$ value increases.

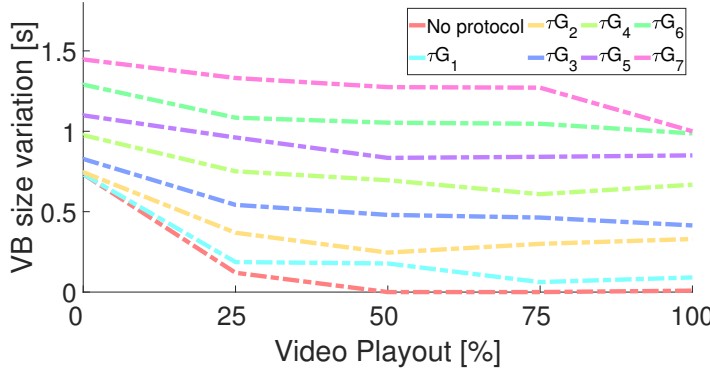

**Figure 9.** Video buffer variations for all scenarios on playout percent function.

In the third experiment, the objective is to pinpoint optimal $\tau$ values to maximize specific criteria. For this experiment, the optimization objective is the performance of ASViS, based on the average amount of sent layer outcomes, using the MM method. It is crucial to note that these optimal values might not always be integers, suggesting a broad range of potential $\tau$ values. The foundational setup for this experiment involved fixed parameters such as an *RTT* of 0.05 s, a packet loss rate of 1/200, and a buffer of 1 s. For BI, a fixed value of $\tau_1$ at 0 s (representing the deadline) was applied. In contrast, for the layers BB, EI, and EB, the values $\tau_2$, $\tau_3$, and $\tau_4$ were adjusted, respectively. Performance was evaluated based on the maximum number of layers transmitted, with a combined limit of 300 layers: 19 for BI, 19 for EI, 141 for BB, and 141 for EB. This evaluation determined the local maxima within each quadrant. For each point, ten pseudo-random seeds were probed, and the average of the maximums was obtained. A consistent search range, spanning from 0 to 8 s, was employed for each $\tau$.

The visualization of these results is provided in a 3D plot, as seen in Figure 10. The axes, labeled as $G_2$, $G_3$, and $G_4$, represent the Gaps (G) or intervals between consecutive $\tau$ values, excluding $\tau_1$. These represent the intervals from the deadline to $\tau_2$, from $\tau_2$ to $\tau_3$, and finally from $\tau_3$ to $\tau_4$. It is crucial to recognize that these intervals are sequential and do not intersect. The overall time span between the deadline and $IG_4$ is the collective sum of $\tau_2$, $\tau_3$, and $\tau_4$. Based on the fixed conditions of the *RTT* and packet loss, optimal performance metrics were identified as $\tau_1$ 0 *RTT*, $\tau_2$ 0 *RTT*, $\tau_3$ 1 *RTT*, and $\tau_4$ 2 *RTT* for the layers BI, BB, EI, and EB, respectively.

For experiment 4, a hypothesis was formulated stating that the gaps in layered-based deadlines (represented by $\tau$ values) are dependent on and proportional to the layer sizes. The video detailed in Table 1 features relatively small base layers and large enhancement layers due to quantization in each layer. To test this hypothesis, the configuration was reversed (i.e., small enhancement layers and large base layers) for testing purposes, and the network conditions from experiment 3 were used. The same three-dimensional maximum value search using the MM method was then performed. The details of each Layer Size Configuration (LSC) can be found in Table 5. This table represents the average accumulated size per layer in a GOP structure of IB × 7. $LSC_1$ corresponds to experiment 3, while $LSC_2$ is the same video from Table 1 but with a quantization of 20 for both BL and EL.

Performance, based on the percentage of sent layer outcomes and derived using the MM method for $LSC_2$, is presented in Figure 11. A preliminary comparison of the two 3D coordinates results, Figures 10 and 11, reveals significant differences in the optimal $\tau$ values between the two LSCs. Detailed coordinates, referencing $\tau_2$, $\tau_3$, and $\tau_4$, are provided in Table 6.

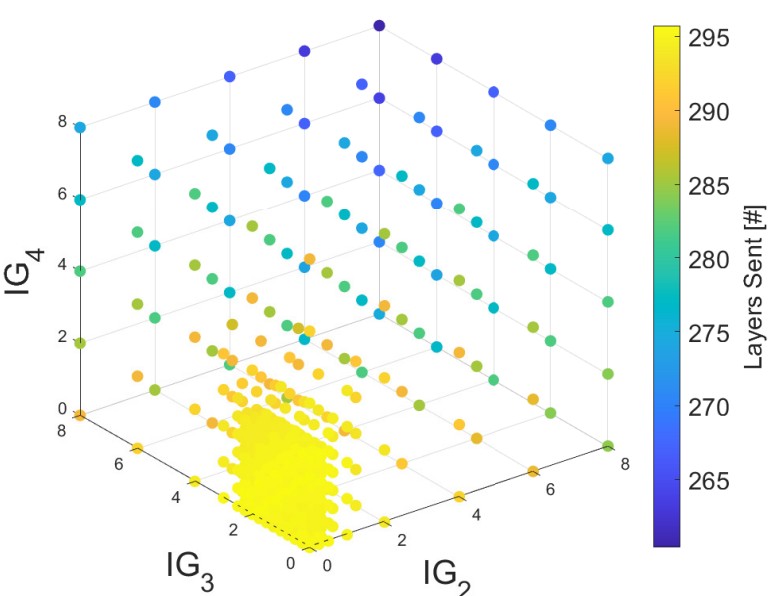

**Figure 10.** Video quality results of MM method for BB, EI, and EB layers for experiment 3.

**Table 5.** Average layer size for both LSCs.

| Layer | $LSC_1$ [kB] | $LSC_2$ [kB] |
| --- | --- | --- |
| BI | 6.97 | 19.31 |
| BB | 36.13 | 23.43 |
| EI | 0.46 | 1.73 |
| EB | 10.29 | 6.66 |

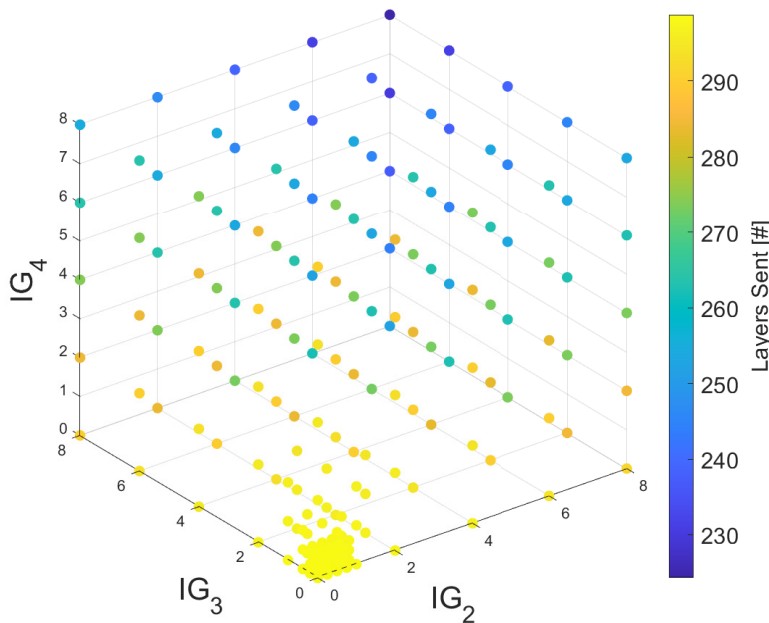

**Figure 11.** Video quality results of MM method for BB, EI, and EB layers for $LSC_2$ of experiment 4.

**Table 6.** Video quality results and $\tau G$ coordinates for different layer size configurations.

| $\tau G$ | $\tau G$ [RTT] | | | Sent Layers [%] | | | |
|---|---|---|---|---|---|---|---|
| | $\tau_2$—BB | $\tau_3$—EI | $\tau_4$—EB | BI | EI | BB | EB |
| $LSC_1$ | 0 | 1 | 2 | 100 | 96.6 | 60.2 | 26.6 |
| $LSC_2$ | 0 | 0.5 | 0.75 | 98.7 | 94.53 | 42.7 | 21.4 |

A thorough examination of the results between $LSC_1$ and $LSC_2$, and the corresponding layer sizes, has revealed a distinct association between the $\tau$ values and the sizes of layers. As depicted in Figure 12, the normalized layer sizes and the $\tau$ values for scenarios $LSC_1$ and $LSC_2$ are juxtaposed. It has been discerned from the blue markers that layer 2 holds a predominant role in both configurations. Conversely, the red markers have indicated that elevated $\tau$ values are associated with more diminutive layers. A noticeable inverse correlation between the layer sizes and $\tau$ values is evident: as the magnitude of a layer amplifies, the optimal $\tau$ for video quality is observed to wane. The pivotal influence of layer sizes on video quality, in the context of $\tau$, is underscored by this relationship.

In the fifth experiment, a comparison was made between the performance of the ABR algorithms, MPC and ASViS, using various parameters. Traditional TCP behavior, which is typical for ABR algorithms, was exhibited under the network conditions for MPC. In contrast, User Datagram Protocol (UDP) with flow control was employed by ASViS, consistent with previous experiments. Network transmission parameters from experiment 3 were replicated, and the $\tau G$ values for the ASViS protocol were derived from the same experiment. A total of 100 simulations were executed for both ABR algorithms, and the results presented were the mean values obtained from all simulations.

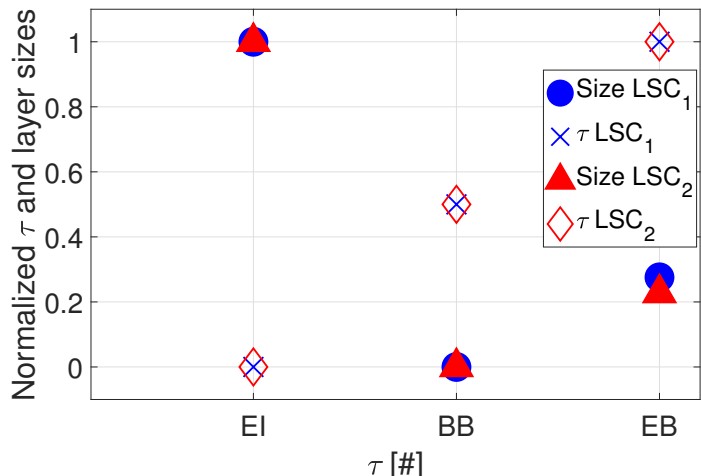

**Figure 12.** Comparison between normalized layer sizes and optimal $\tau$ ranges.

A chunk size of 4 GOP was designated for MPC, equivalent to 32 frames or roughly 1 s of playout, given a frame rate of 30 fps. Four bitrate levels, labeled as $R$, were used, each integrating various layers. Comprehensive details of each $R$ can be accessed in Table 7. An average across all chunks served as a reference for AVQ of Y-PSNR in this MPC implementation, accompanied by a chunk size ($d(R)$) specified in bits. The initial $T_s$ spanned from 0.1 s to 5 s, incremented by 0.2 s. The estimated throughput ($C$) was calculated as the harmonic mean throughput of the previous five chunks, with weight assignments of $\lambda = 1$, $\beta = 3000$, and $\beta_s = 3000$, aligning with the recommendations in [49].

As illustrated in Figure 13, the relationship between the VB size variation and video playout was mapped. Results indicated that a mean VB size of 1.5 s with a standard deviation of 0.5 s was achieved by MPC, whereas ASViS achieved a mean of 0.45 s with a deviation of 0.08 s. Notably, greater stability after 20% of video playout was observed with ASViS, despite its reduced mean VB size.

**Table 7.** Chunk properties of each bitrate level R.

| | Layer | | | | Metrics | |
|---|---|---|---|---|---|---|
| | **BI** | **BB** | **EI** | **EB** | **AVQ Y-PSNR [dB]** | $d(R)$ **[Mb]** |
| $R_1$ | X | X | - | - | 28.4 | 0.32 |
| $R_2$ | X | X | X | - | 37.7 | 1.45 |
| $R_3$ | X | X | - | X | 31.0 | 2.70 |
| $R_4$ | X | X | X | X | 41.6 | 3.83 |

Figure 14 portrays the anticipated bitrate distribution for both MPC and ASViS. This visualization aids in comprehending the adaptability of each algorithm to network conditions. An in-depth analysis confirmed that the bitrate estimated distribution for ASViS predominantly focuses on values higher than those for MPC, a pattern that contrasts with its mean and standard deviation provided in Table 8.

Subsequent evaluations explored video quality throughout the entire video playback. The Y-PSNR findings are presented in Figure 15, whereas VMAF results can be found in Figure 16. Both algorithms exhibited comparable video quality from frames 40 to 130. However, at the commencement and conclusion of the video playback, the quality of MPC was observed to be inferior to that of ASViS. This discrepancy is attributed to the conservative approach of MPC and its slower adaptation to network conditions compared to ASViS.

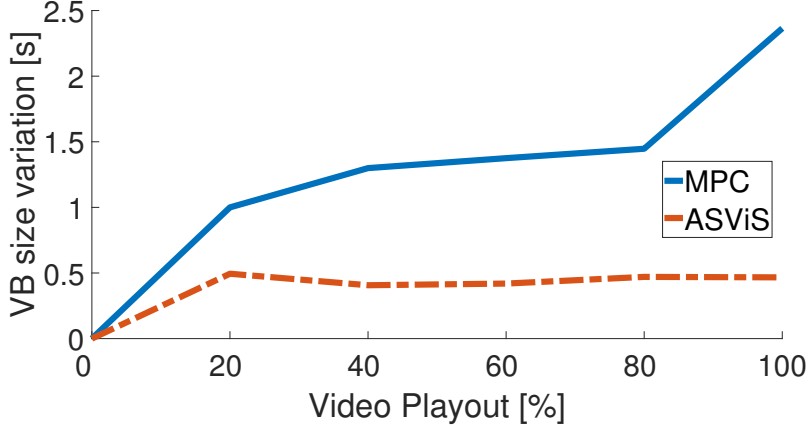

**Figure 13.** VB size comparison through video playout for ASViS and MPC.

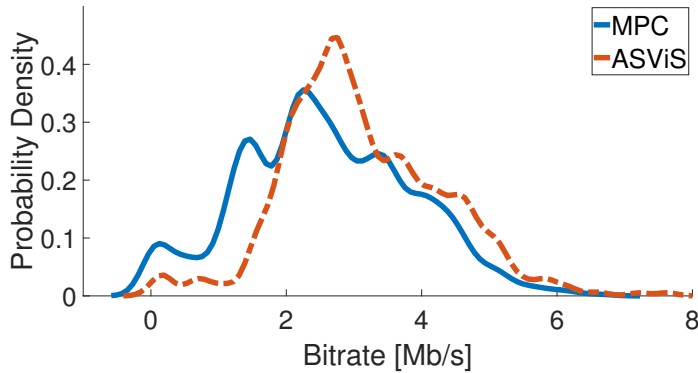

**Figure 14.** Estimated bitrate distribution for ASViS and MPC.

Furthermore, a noteworthy distinction between the results of frames 74 and 84 was observed in Figure 16, owing to the capability of VMAF to discern details overlooked by Y-PSNR. Detailed analysis revealed that frames 74 and 84, corresponding to a B4 frame as cross-referenced with Figure 1b, were dropped by ASViS.

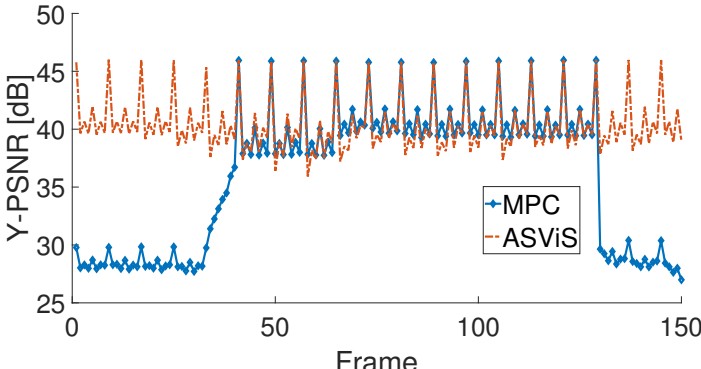

**Figure 15.** Video quality behavior of Y-PSNR for ASViS and MPC.

The video quality metrics, presented in Figures 15 and 16, were analyzed using a distinctive approach involving a boxplot and its corresponding histogram. The boxplot delineates the median, the 25th percentile at the bottom, and the 75th percentile at the top. Whiskers extend to cover values not deemed outliers. Notably, since no outliers were observed, there are no '+' markers. The results for Y-PSNR and VMAF are displayed in Figures 17 and 18, respectively. Although Y-PSNR and VMAF are founded on different philosophies, both metrics consistently show the superiority of ASViS over MPC. For a comprehensive understanding of the mean and deviation values of Y-PSNR, VMAF, and bitrate, one can refer to Table 8. It becomes evident that, in terms of VMAF, Y-PSNR, and bitrate, the ASViS algorithm holds an edge over MPC.

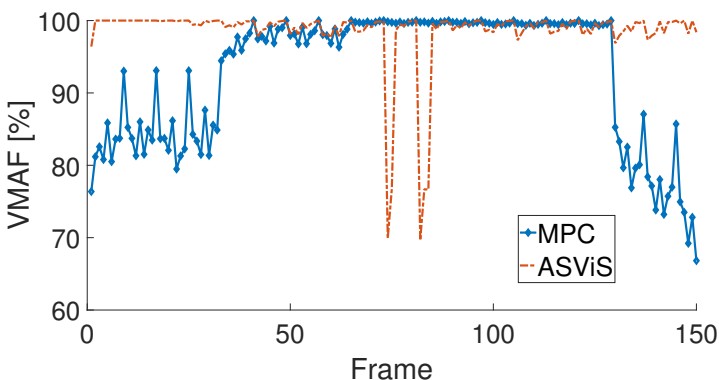

**Figure 16.** Video quality behavior of VMAF for ASViS and MPC.

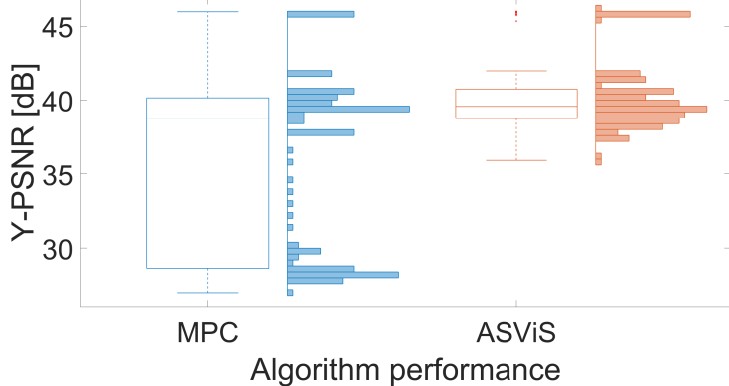

**Figure 17.** Boxplot and histogram for Y-PSNR comparison of ASViS and MPC.

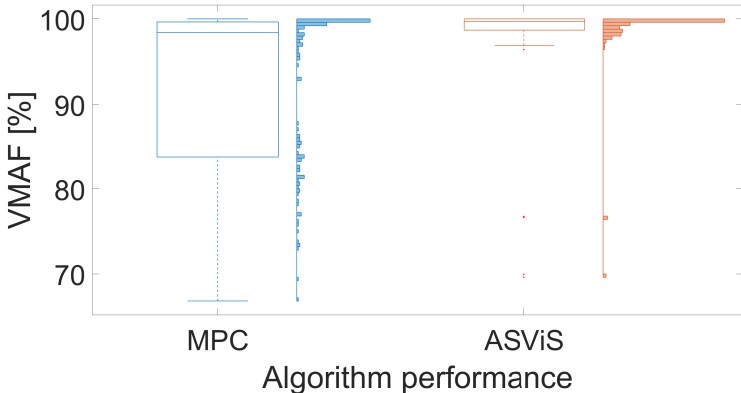

**Figure 18.** Boxplot and histogram for a VMAF comparison of ASViS and MPC.

**Table 8.** Detailed results of experiment 5.

|  | ASViS | | MPC | |
| --- | --- | --- | --- | --- |
|  | **Mean** | **Std** | **Mean** | **Std** |
| Y-PSNR [dB] | 41.1 | 2.4 | 39.2 | 6.0 |
| VMAF [%] | 98.5 | 4.7 | 92.8 | 9.0 |
| Bitrate [Mb/s] | 3.1 | 1.2 | 2.6 | 1.3 |

*5.2. Discussion*

The initial experiment demonstrates a close alignment between theoretical and observed behaviors of ASViS, with mean absolute percentage errors ranging between 6.5% and 14.2%. These findings substantiate the capability of the model to predict the experimental behavior of ASViS, thus eliminating the need for time-consuming emulations or tests.

Outcomes of the second experiment indicate superior video quality in a no-protocol scenario (default SVC) compared to any $\tau$ gap configuration scenario. However, this scenario exhibits video stalls, with rebuffering incidents accounting for over half the video playback duration. In contrast, all $\tau G_x$ scenarios display variations in video buffering without complete depletion. Video quality can vary based on configuration, offering insights into how ASViS modulates video quality to ensure uninterrupted playback.

The third experiment leverages the MM method to discern specific $\tau$ coordinates that yield improved video quality under given network conditions. It reveals that optimal video quality can span a broad range of $\tau$ values and might be envisioned as a multi-dimensional solution. In the fourth experiment, a pronounced inverse correlation is found between the $\tau$ values and layer sizes. This discovery is pivotal, suggesting that service providers can sidestep the task of pinpointing optimal values for individual videos. They can instead modify layer deadlines in proportion to layer sizes, ensuring nearly optimal outcomes.

Findings of the fifth experiment spotlight the superiority of ASViS over MPC in multiple aspects. ASViS records a 5.8% and 4.6% higher Y-PSNR and VMAF, respectively, than MPC. Moreover, ASViS displays a consistently compact video buffer size compared to MPC. It also boasts more efficient network utilization, with a bitrate surpassing that of MPC, and steadier performance, reflected in a reduced standard deviation. In conclusion, these promising results position ASViS as a skillful ABR algorithm.

While the potential of ASViS has been demonstrated in specific contexts through our experiments, its performance in broader or more challenging scenarios remains an intriguing area of speculation. In 5G mobile networks, known for high download speeds and reduced latency, it is hypothesized that ASViS could deliver ultra-high-definition video streaming with minimal interruptions. Conversely, in regions with underdeveloped internet infrastructure or in satellite networks with inherent high latency, challenges may arise regarding how ASViS adapts video flow and manages fluctuations. It might be

theorized that smooth but lower-quality video playback could prevail. Further research is warranted in these areas. Future directions are aimed at exploring and validating these scenarios to ensure the resilience of ASViS beyond ideal conditions. For future directions, it is proposed that more exhaustive tests be conducted, involving videos with varying resolutions and different frame rates. Additionally, it has been identified that simulations need to be carried out under realistic network conditions, such as those of mobile networks.

## 6. Conclusions

In the current landscape, robust ABR algorithms are vital to meet the vast demands of video streaming on global networks. These algorithms must ensure superior video quality, prevent stalls, and adapt swiftly to network fluctuations. The introduction of the novel concept of ASViS has been demonstrated through experiments to efficiently control video stream flow in bandwidth-limited scenarios. When SVC integrates with ASViS, it not only enhances the user QoE but also reduces network congestion. Such improvements arise from aligning the encoding rate with available throughput and discarding less critical layers.

Based on the tests conducted and the information gathered, it is suggested that the integration of DASH with ASViS enables the server to better adapt to real-time conditions, transmitting only the layers that the client can handle according to their bandwidth capacity. The ASViS proposal is found to be dynamically adaptable to network conditions and device capabilities, allowing for a more efficient use of bandwidth and an enhanced user experience, reducing start-up times and rebuffering. Furthermore, in terms of storage, it is necessary for the server to store only a single version of the scalable video. DASH is agnostic in terms of encoding, so the integration of SVC does not necessitate significant changes to the existing DASH infrastructure. Likewise, the new generation of video codec solutions, such as H.265, Video Predictor 9, and AOMedia Video 1, inherently possess the scalable encoding model.

In the conclusions drawn, ASViS is portrayed as a transformative solution to ABR algorithms, with inherent limitations of DASH being addressed. Through the experiments conducted, the proficiency of ASViS in managing video stream flow, enhancing user QoE, and mitigating network congestion was validated. A noteworthy alignment was observed between theoretical and experimental behaviors, as evidenced by the mean absolute percentage errors, further emphasizing the reliability of ASViS. With its dynamic adaptability, bandwidth utilization was optimized, video quality was elevated, and interruptions were minimized, setting it apart from algorithms such as MPC. In the tested scenarios, the efficacy of ASViS was demonstrated, marking a significant advancement in video streaming.

**Author Contributions:** E.P.-A. contributed on the design, implementation, data collection, and analysis of the proposed work. C.E. contributed on the design, implementation, and analysis of the proposed work. A.S. contributed on the design of the proposed work. M.O. contributed on the analysis of the proposed work. All authors have read and agreed to the published version of the manuscript.

**Funding:** This work is partially funded by the Comision Nacional de Investigacion Cientifica y Tecnologica (CONICYT) Doctorate Scholarship 2018-21181143, Fondo Nacional de Desarrollo Cientifico y Tecnologico (FONDECYT) Chile Grant 1210031, and the Advanced Center for Electrical and Electronic Engineering (AC3E), Agencia Nacional de Investigacion y Desarrollo (ANID) Basal Project FB0008.

**Data Availability Statement:** Videos are found at [56], this is a public site not managed by the authors. Please contact the correspondence author for any other data of interest.

**Conflicts of Interest:** The authors declare no conflict of interest.

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
