# Peer review of "Adaptive Scalable Video Streaming (ASViS): An Advanced ABR Transmission Protocol for Optimal Video Quality"

_electronics, doi:10.3390/electronics12214542_

Round 1

Reviewer 1 Report

Comments and Suggestions for Authors

Introduction: The introduction sets the stage well by explaining the importance of video streaming and the challenges associated with it. However, you might consider providing some statistics or references to support the claim that multimedia video streaming is the most used data consumption service on the internet.

DASH: When mentioning DASH, you could briefly explain what it stands for (Dynamic Adaptive Streaming over HTTP) for the benefit of readers who may not be familiar with the term.

ASViS: Provide a brief explanation of the acronym ASViS when it's first introduced (Adaptive Scalable Video Streaming) to ensure clarity.

Technical Details: It's important to provide a bit more detail on how ASViS achieves its goals. You mention "scalable video coding" and "deadline-based criteria," but a sentence or two on how these work would be beneficial for understanding.

Comparison to TCP: The comparison to TCP is a good point. Consider expanding on this a bit to highlight the key advantages of ASViS over TCP in the context of video streaming.

Compliance with RFC 8085: It's mentioned that ASViS is compliant with RFC 8085, which is excellent. However, it might be beneficial to briefly explain what RFC 8085 is and why compliance with it is important.

Scalability: While you mention that ASViS has scalability features, consider providing a brief explanation of what scalability means in the context of video streaming and why it's advantageous.

Theoretical Model: Highlight the key findings or predictions from the theoretical model. This will give the reader a sense of what to expect from the rest of the article.

Parameters: Mention the specific parameters you will be discussing and how they affect ASViS' performance. This provides a roadmap for what readers can expect in the following sections.

Conclusion: Consider adding a sentence or two summarizing the main takeaways or benefits of ASViS

Author Response

Introduction: The introduction sets the stage well by explaining the importance of video streaming and the challenges associated with it. However, you might consider providing some statistics or references to support the claim that multimedia video streaming is the most used data consumption service on the internet.

We appreciate the acknowledgement that the introduction sets the stage well. We are committed to make the document as clear and self-contained as possible while attempting to be concise. Some statistics and references that support our claims will be added.

This is the inserted text, which is highlighted light yellow in the main document.

“Some had predicted that by 2020, video traffic would account for 82\% of global online traffic \cite{Meligy}. Contrary to these predictions, it is reported in \cite{Sandvine} that in the first half of 2022, video streaming constituted 65.93\% of total internet traffic, with Netflix and YouTube contributing 13.74\% and 10.51\%, respectively. Furthermore, a 23\% increase in total traffic volume was observed in 2022 compared to 2021, attributed to the significant growth of various streaming services. Current data regarding video streaming consumption on mobile networks indicates that 71\% of the traffic was video-related \cite{Ericsson}. Projections suggest that this figure is expected to reach 80\% by 2028. Such statistics underscore the undeniable growth trend.”

DASH: When mentioning DASH, you could briefly explain what it stands for (Dynamic Adaptive Streaming over HTTP) for the benefit of readers who may not be familiar with the term.

This is a good observation. The term DASH was first mentioned in the abstract and therefore the acronym meaning was mentioned there, since that was its first use. Nevertheless, it is always uncertain if when mentioned in the abstract it should be also mentioned again in the first use of the main document as the readers rarely go back to the abstract to check for terms. To make it easier for the readers to find the terms we will explain the acronyms both in the abstract and in the first use of the core of the document.

We have also added the sentence “Dynamic Adaptive Streaming over HTTP (DASH) is an open standard for transmitting video that adapts to both the network and device.” to make the document a little more self-contained.

The inserted text, is highlighted light yellow in the main document.

ASViS: Provide a brief explanation of the acronym ASViS when it's first introduced (Adaptive Scalable Video Streaming) to ensure clarity.

Thank you for your feedback and comments. We would like to point out that the acronym "ASViS" (Adaptive Scalable Video Streaming) was indeed explained when it was first introduced in the abstract: "This study introduces the Adaptive Scalable Video Streaming (ASViS) protocol as a solution." Nevertheless, similar to the previous terms, we understand the importance of ensuring clarity for readers, and we made sure to provide the full term alongside the acronym when first mentioned, both in the abstract and in the core of the document. 

Technical Details: It's important to provide a bit more detail on how ASViS achieves its goals. You mention "scalable video coding" and "deadline-based criteria," but a sentence or two on how these work would be beneficial for understanding.

Thank you for your insightful feedback regarding the need for further elaboration on "scalable video coding" and "deadline-based criteria". Your observation is highly valuable, and we have taken steps to incorporate additional technical details in the manuscript to provide a clearer understanding of how ASViS achieves its goals. The inserted text, is highlighted light yellow in the main document.

“At its core, ASViS leverages Scalable Video Coding (SVC) to partition video data into base and enhancement layers, with selective transmission based on layer-discarding policies. This differentiation is orchestrated by the application of a deadline-sensitive criteria. Specifically, each layer possesses a distinct "curfew", determining its transmission or omission based on the nearing of its playback deadline. For instance, as deadlines approach, higher-layer data might be excluded, ensuring the arrival of the base layers, which are more crucial for video coherence, are transmitted.”

Comparison to TCP: The comparison to TCP is a good point. Consider expanding on this a bit to highlight the key advantages of ASViS over TCP in the context of video streaming.

This is a great suggestion and we appreciate the input. We have added a paragraph explaining the contrast between ASViS, which uses UDP, and other protocols that are based on TCP. The paragraph is below and it is highlighted light yellow in the main document:

“Compared to traditional video streaming methods based on TCP, which often rely heavily on retransmissions and sequence acknowledgments, ASViS, with its refined layer-discarding policy and strategic use of UDP, offers a more efficient and adaptable approach.”

Compliance with RFC 8085: It's mentioned that ASViS is compliant with RFC 8085, which is excellent. However, it might be beneficial to briefly explain what RFC 8085 is and why compliance with it is important.

As suggested, in the interest of the readers we have explained briefly RFC 8085 in our manuscript. Due to the relation to the previous question, we have bundled the explanation in the same paragraph as the ASCiS vs TCP answer. So immediately following those sentences, in the same paragraph, we have added  highlighted light yellow:

“This methodology not only reduces overhead but also aligns closely with the RFC 8085 guidelines. These guidelines are a series of recommendations on UDP usage and best practices designed to prevent network congestion. By complying with these guidelines, stability and efficiency in network operations are ensured. The adherence of ASViS to this standard is underscored by its significance.”

Scalability: While you mention that ASViS has scalability features, consider providing a brief explanation of what scalability means in the context of video streaming and why it's advantageous.

Thanks for this observation, the term scalability can mean different things in different contexts. We have attempted to clarify what is meant by scalability in the context of ASViS. The answer is below and included and highlighted light yellow in the main document.

“Scalability is an important factor in the growth of any technology. When referring to ASViS, scalability is characterized by several factors: 1. Since ASViS is based on SVC, which itself is part of H.264 (and newer), it can easily evolve to higher resolutions (8K, 16K, etc.). 2. SVC does not have a fixed amount of layers, to prove our concept this work uses four SVC layers, but more can be used. As technology progresses and resolutions increase, it is beneficial to also increase the amount of SVC layers used, since it would create more granularity (i.e., more quality options to choose from). There are other aspects not covered in this work, in which ASViS can scale (such as GOP size, block compression size, etc.) that are inherent to the compression standards. All these degrees of scalability allow ASViS to maintain itself relevant in the future.”

Theoretical Model: Highlight the key findings or predictions from the theoretical model. This will give the reader a sense of what to expect from the rest of the article.

This is a great suggestion. To land this theoretical topic a little softer for the reader and give a better intuition of what to expect we have included this explanation highlighted light yellow in the manuscript:

“Given the inherent variability of the practical experiments due to packet losses and the intricate characteristics of each frame, along with the nuances of the various layers and complexities associated with network conditions in empirical analyses, the accuracy of the theoretical model is underscored. It is noteworthy that high fidelity is achieved by the theoretical model, aligning well with the experimental results. This contributes to a reduction in the need for exhaustive and computationally demanding empirical tests, thereby facilitating more streamlined behavioral predictions based on the theoretical model. It is important to stress that this does not replace empirical benchmarks, but can significantly reduce the amount of energy, resources, and time devoted to more practical tests.”

Parameters: Mention the specific parameters you will be discussing and how they affect ASViS' performance. This provides a roadmap for what readers can expect in the following sections.

We are committed to provide clear explanations of all parameters and how they influence the behavior of ASViS. It would be of interest to us to know if any parameters in particular are not sufficiently clear in the current manuscript. We have included this paragraph to pave the way into the current explanations in hope to make it clearer of what to expect. Hopefully, we’ll have another opportunity to clarify any parameters and its effects, which the reviewers find that are lacking detail or are unclear on how it affects the behavior of ASViS. The added text highlighted light yellow  is the following:

“The performance of ASViS is influenced by several specific parameters, which will be elaborated upon in the subsequent sections. Notably, network conditions, such as Round Trip Time (RTT), packet loss, and buffer conditions, have been identified. Additionally, variations in \tau parameters across different layers and alterations in the size of each layer have been observed to impact performance. A thorough examination of these factors and their implications on ASViS functionality is included in the following discussions.”

Conclusion: Consider adding a sentence or two summarizing the main takeaways or benefits of ASViS

Thank you for the valuable suggestion. It is extremely important for us that the readers understand the benefits of ASViS in a centric, concise, and clear manner. For this reason we have summarized all conclusions and contributions in a single closing paragraph where everything converges comprehensively. Here is the added highlighted paragraph that brings everything together:

“In the conclusions drawn, ASViS is portrayed as a transformative solution to ABR algorithms, with inherent limitations of DASH being addressed. Through the experiments conducted, the proficiency of ASViS in managing video stream flow, enhancing user QoE, and mitigating network congestion was validated. A noteworthy alignment was observed between theoretical and experimental behaviors, as evidenced by the mean absolute percentage errors, further emphasizing the reliability of ASViS. With its dynamic adaptability, bandwidth utilization was optimized, video quality was elevated, and interruptions were minimized, setting it apart from algorithms like MPC. In the tested scenarios, the efficacy of ASViS was demonstrated, marking a significant advancement in video streaming.”

Reviewer 2 Report

Comments and Suggestions for Authors

After reviewing the article Adaptive Scalable Video Streaming (ASViS): An Advanced ABR Transmission Protocol for Optimal Video Quality, I consider that it is based on a methodology that is adequate to the objectives stated in the scientific article. In addition, after reviewing the literature on which the article is based, I consider that it is current and reliable to frame the article at a scientific level. Likewise, I consider that the conclusions cover the objectives and clearly expose the results obtained from the research. Finally, the future conclusions shed light on where the research is headed and I consider them to be very opportune.

Author Response

Thank you for your comments. We are very glad to hear that you found the manuscript clear, relevant, and self-contained.

Reviewer 3 Report

Comments and Suggestions for Authors

Very well-written paper with detailed explanations of the study design and the results obtained from different experiments. I liked how various experiments were conducted to evaluate specific aspects of ASViS since it aims to solve a 'multi-layered' problem.

I have no problems in recommending this work for publication as is. Yet, since the content is of a high quality, perhaps a short discussion of how ASViS might perform in different scenarios would make it even better and more comprehensive. The authors conclude by stating that there are some obvious future paths for their research (introducing more variety in video test sequences and experimenting in more real-life scenarios), but I think a short discussion of what could be expected in those experiments would be beneficial for this paper.

Author Response

Very well-written paper with detailed explanations of the study design and the results obtained from different experiments. I liked how various experiments were conducted to evaluate specific aspects of ASViS since it aims to solve a 'multi-layered' problem.

I have no problems in recommending this work for publication as is. Yet, since the content is of a high quality, perhaps a short discussion of how ASViS might perform in different scenarios would make it even better and more comprehensive. The authors conclude by stating that there are some obvious future paths for their research (introducing more variety in video test sequences and experimenting in more real-life scenarios), but I think a short discussion of what could be expected in those experiments would be beneficial for this paper.

We appreciate the suggestion and completely agree that the manuscript would benefit significantly from depicting other scenarios, such as common real-life scenarios, and discuss how ASViS might perform. The following paragraph has been added to the manuscript in the discussion section (highlighted in light green):

“While the potential of ASViS has been demonstrated in specific contexts through our experiments, its performance in broader or more challenging scenarios remains an intriguing area of speculation. In 5G mobile networks, known for high download speeds and reduced latency, it is hypothesized that ASViS could deliver ultra-high-definition video streaming with minimal interruptions. Conversely, in regions with underdeveloped internet infrastructure or in satellite networks with inherent high latency, challenges may arise regarding how ASViS adapts video flow and manages fluctuations. It might be theorized that smooth but lower-quality video playback could prevail. Further research is warranted in these areas. Future directions are aimed at exploring and validating these scenarios to ensure the resilience of ASViS beyond ideal conditions. For future directions, it is proposed that more exhaustive tests be conducted, involving videos with varying resolutions and different frame rates. Additionally, it has been identified that simulations need to be carried out under realistic network conditions, such as those of mobile networks.”

Reviewer 4 Report

Comments and Suggestions for Authors

This work introduces the ASViS protocol, which adjusts the bitrate according to the available bandwidth in the network. The manuscript is well structured and written.
Five experiments have been conducted in order to evaluate the protocol performance, well summarized in the "Discussion" section. The results are statistically-sound.
According to the results, the theoretical model of the ASViS protocol match with the experiments. In terms of Y-PSNR, the second experiment apparently shows that a better performance is achieved without the ASViS protocol, however it avoids rebuffering, that is, there are no stalls. Two metrics have been used to compare the video quality of ASViS and MPC: PSNR and VMAF, achieving better results with ASViS, and, in addition, it uses less bandwidth, that is, a more efficient of the network is done.

The H.264 video coding standard is a bit old, although it is still used; other more recent ones have emerged, as mentioned in the article itself (HEVC, etc.). Also, I cannot find in the manuscript the Y-PSNR value for the original encoded video sequence. Finally, the software used form the simulations is not explained, I think.

About ACRONYMS: they all always well defined at their first usage, although some of the acronyms defined in the abstract are not defined again in the main text, as: TCP (line 28), DASH (28 and perhaps also in 192), UDP (40), SVC (43).

Some minor typing mistakes or suggestions:
- line 200: "framerate" -> "frame rate"
- line 490: "It's"      -> "It is"

About FIGURES, all of them have enough quality (resolution) and looks good. Only in Fig.12, the symbols of the legend are too small, and on the graph, the red squared markers overlapped too much the blue rounded ones; perhaps a red triangle could looks better than a red squared.

The BIBLIOGRAPHY seems to be complete and updated.
[46] "Mpc" -> "MPC"

Author Response

This work introduces the ASViS protocol, which adjusts the bitrate according to the available bandwidth in the network. The manuscript is well structured and written.

Five experiments have been conducted in order to evaluate the protocol performance, well summarized in the "Discussion" section. The results are statistically-sound.

According to the results, the theoretical model of the ASViS protocol match with the experiments. In terms of Y-PSNR, the second experiment apparently shows that a better performance is achieved without the ASViS protocol, however it avoids rebuffering, that is, there are no stalls. Two metrics have been used to compare the video quality of ASViS and MPC: PSNR and VMAF, achieving better results with ASViS, and, in addition, it uses less bandwidth, that is, a more efficient of the network is done.

We greatly appreciate your generous evaluation of the overall state of the paper.

The H.264 video coding standard is a bit old, although it is still used; other more recent ones have emerged, as mentioned in the article itself (HEVC, etc.).. 

Thank you for your observation. This is very true, but we believe that we have always mentioned that the proposed solution works in H.264 or newer. We will double check to make sure of this. If the comment is a result of the background coverage, there are a few reasons for including H.264 in the background opposed to newer versions. The first reason is that the inception of SVC was in H.264, and it best describes the mechanisms and purpose. Since then it has evolved to H.265 and H.266 including more dimensions (temporal (frame rate), spatial (resolution), SNR, color gamut, dynamic range differences, stereo/multiview coding, panoramic formats, and still-picture coding), but there is less detail on how SVC itself works, as it relies on the reader to be familiar with H.264. The second reason is that H.264 is the most used video codec as of April 15, 2023 (The 3 Most Popular Video Codecs Used Worldwide (free-codecs.com) mainly because many of the major streaming services, such as YouTube, Netflix, and Amazon Prime Video, still use it. We hypothesize this could be the result of compromising compatibility with most devices and choosing the newest codec possible. The third reason is that we had the whole experimental setup working in matlab with the H.264 compiler. In this work, the main purpose is to prove the advantages of using the SVC extension in a cross-layer solution with the application layer to determine which SVC layers to transmit using the deadline-based criteria. Without any loss of generality this shows it can be done, meaning that H.265 and H.266 can also accomplish similar results, and probably with even better results due to the level of granularity than can be achieved by breaking down the video in more dimensions. For now we leave this as future work. What is important for now is to create awareness of the advantages of using the SVC extension, which we believe is underutilized. We truly believe this work has the potential to inspire and steer more work in this direction.

Also, I cannot find in the manuscript the Y-PSNR value for the original encoded video sequence.

Thank you for your feedback, and keen observation. The Y-PSNR information was added to Table 1 along with the VMAF.

Finally, the software used form the simulations is not explained, I think.

Thank you for suggesting this as including these details can help others understand better our methodology and perhaps even use the same technique. We have added a paragraph explaining the software used and other details, like how to link the software. This is the text added in the manuscript, you can find it easily by searching the light red highlighted region.  

“The ASViS algorithm was implemented using MATLAB and JSVM. Different network conditions were simulated based on the layers’ information from the trace file of JSVM, determining whether packets were transmitted or discarded. Furthermore, MATLAB was employed in conjunction with JSVM for tasks associated with video encoding/decoding. Additionally, the results were organized and processed using MATLAB. JSVM works using the operating system command prompt. It is easy to link JSVM to MATLAB using the matlab {\textit{system}} command.”

About ACRONYMS: they all always well defined at their first usage, although some of the acronyms defined in the abstract are not defined again in the main text, as: TCP (line 28), DASH (28 and perhaps also in 192), UDP (40), SVC (43).

We completely agree and this was also mentioned by another reviewer. This is sometimes considered ‘double defining’ and unnecessary, though it is our opinion that sometimes readers don’t check the abstract for definitions and it makes it more difficult to find their meanings. Double defining is sometimes necessary for a better reading experience. Therefore we have made the appropriate changes and defined it in both the abstract and the first time it is used in the core manuscript. 

Some minor typing mistakes or suggestions:

- line 200: "framerate" -> "frame rate"

- line 490: "It's"  -> "It is"

Thank you for pointing out these details. We always attempt to write in a formal tone and avoid contractions like it’s, isn’t, can’t, doesn’t, etc. We have checked the entire document again to avoid these, keep in mind there is always a chance we missed something, we checked for the most common ones. The frame rate was a typo. We always try, time-permitting, to double and even triple check all spelling and grammar issues, but some technical terms might have slipped through.

About FIGURES, all of them have enough quality (resolution) and looks good. Only in Fig.12, the symbols of the legend are too small, and on the graph, the red squared markers overlapped too much the blue rounded ones; perhaps a red triangle could looks better than a red squared.

We are glad to hear that all the figures with the exception of 12, are clear and readable. Regarding, specifically, to Figure 12, we have made the necessary adjustments to enhance the clarity of the chart. As suggested, we have replaced the red squared markers with red triangles to avoid overlapping with the blue rounded ones. Additionally, the symbols in the legend have been resized for better visibility. We appreciate your keen observation and hope these changes improve the readability of the figure.

Reviewer 5 Report

Comments and Suggestions for Authors

I appreciate the documentation made by the authors of this article proposal by means of the bibliographic references invoked in this regard, usually very recent and related to the research area followed and in-depth.

I also appreciate the applied component of the article proposal through which the models developed previously from a theoretical point of view are empirically validated through multiple experiments and through the results obtained.

I also appreciate the reality and feasibility of the conclusions drawn following the testing and validation of the research results undertaken by the authors of this article proposal.

Therefore, taking into account the above, I recommend the publication of this article proposal.

Author Response

I appreciate the documentation made by the authors of this article proposal by means of the bibliographic references invoked in this regard, usually very recent and related to the research area followed and in-depth.

I also appreciate the applied component of the article proposal through which the models developed previously from a theoretical point of view are empirically validated through multiple experiments and through the results obtained.

I also appreciate the reality and feasibility of the conclusions drawn following the testing and validation of the research results undertaken by the authors of this article proposal

Therefore, taking into account the above, I recommend the publication of this article proposal.

Thank you for your thorough review and positive feedback. We are heartened to learn that you found the manuscript to be clear, pertinent, and comprehensive. Your recognition of the depth of our bibliographic references, the empirical validation of our models, and the feasibility of our conclusions is greatly appreciated. We value your recommendation for the publication of this article.

Reviewer 6 Report

Comments and Suggestions for Authors

This paper presents a cross-layer solution-the Adaptive Scalable Video Streaming (ASViS) protocol-which offers a promising approach to solving the challenges faced by DASH by combining scalable video coding, flow-control User Datagram Protocol (UDP), and deadline-driven standards.

In additional, the ASViS also achieves the highest image quality per frame, consistent with Scalable Video Coding (SVC) and the available data layers.

However, the paper needs to address a few points before being considered for publication.

1.        In Section VI, the authors should make it clear what the difference is between the theoretical arrival time and the experimental arrival time in Figure 6? Why does $\tao G_7 $ have the largest gap between theoretical and experimental?

2.        it is recommended to further illustrate the complexity comparison of ASViS and other related methods.

3.        In Section IV, the author introduces the MM method to optimize parameter tao. If this method is original and how complex is it?

4.        A careful check of the overall paper is advised to remove the typos and grammatical errors.

Comments on the Quality of English Language

A careful check of the overall paper is advised to remove the typos and grammatical errors.

Author Response

This paper presents a cross-layer solution-the Adaptive Scalable Video Streaming (ASViS) protocol-which offers a promising approach to solving the challenges faced by DASH by combining scalable video coding, flow-control User Datagram Protocol (UDP), and deadline-driven standards.

In addition, the ASViS also achieves the highest image quality per frame, consistent with Scalable Video Coding (SVC) and the available data layers.

In Section VI, the authors should make it clear what the difference is between the theoretical arrival time and the experimental arrival time in Figure 6? 

Thank you for your observation regarding Figure 6. The overall difference is that the theoretical trace is obtained using a simulation where the behavior conforms to the equations derived in this work and are an important contribution. The experimental case is using an uncompressed video file, compressing using JSVM, splitting into layers and transmitting over an emulated network using matlab, then reversing the whole process to restore the video. 

The difference between the theoretical and experimental arrival times stems from how packet sizes are estimated:

  • The theoretical model relies on the Estimated Packet Size (EPS) derived from a trace file created for JSVM, providing an average layer size.
  • The experimental model, on the other hand, takes into account the actual size of each layer, influenced by its content and classification as either BL or EL and as I or B.

This leads to consistent sizes in the theoretical model and varying sizes in the experimental one. To further clarify these distinctions, we have added 2 paragraphs to the manuscript (highlighted in light orange):

“In the theoretical model, the determination of which packets are discarded is based on the estimated packet size (EPS) and network conditions as utilized by the ASViS algorithm. The EPS values are informed by the mean layer size information derived from a trace file, which is created for JSVM. The size in MSS for each layer is presented in Table \ref{EPS}. For the experimental model, the actual size of each layer, influenced by its content, whether it is BL or EL, and its classification as I or B, is utilized. This results in a variety of sizes for each layer.”

“Given the inherent variability of the practical experiments due to packet losses and the intricate characteristics of each frame, along with the nuances of the various layers and complexities associated with network conditions in empirical analyses, the accuracy of the theoretical model is underscored. It is noteworthy that high fidelity is achieved by the theoretical model, aligning well with the experimental results. This contributes to a reduction in the need for exhaustive and computationally demanding empirical tests, thereby facilitating more streamlined behavioral predictions based on the theoretical model. It is important to stress that this does not replace empirical benchmarks, but can significantly reduce the amount of energy, resources, and time devoted to more practical tests.”

Why does $\tao G_7 $ have the largest gap between theoretical and experimental?

This is a very good observation of Fig. 6. There are various reasons that can cause this discrepancy.  The main reasons that could cause this notable deviation between theoretical and experimental results for \tau G_7 are the average vs actual packet size, video complexity variations.These factors have been detailed in the manuscript for clarity. It is worth mentioning that the video length is very short, which allows the scale of the graph to include more detail. Nevertheless, the discrepancy between the experimental and theoretical traces is about 300-400 ms. After around frame 50, the traces stabilize and their error will not continue to grow, as both the theoretical and experimental protocol behaviors are attempting to get as close as possible to the deadline without exhausting the buffer. 

Also we have added 1 paragraph to the manuscript (highlighted in light orange):

“A significant gap between theoretical and experimental results is observed for $\tau G_7$. This discrepancy can be attributed to various factors, including differences in average vs. actual packet size and video complexity variations.”

it is recommended to further illustrate the complexity comparison of ASViS and other related methods.

The complexity of ASViS, when compared to other methods, was evaluated to ascertain its practical applicability. Video data was partitioned by ASViS using Scalable Video Coding (SVC). Although the initial compression and layering of uncompressed videos were recognized as time-intensive and potentially unsuitable for real-time operations, it was noted that these processes could be completed offline. Once converted to SVC, layer-based discarding was found to be simple and swift. In contrast, cumbersome overheads were often associated with traditional methods like DASH.

In Section IV, the author introduces the MM method to optimize parameter tao. If this method is original and how complex is it?

This is a very good observation. Though we have never seen this technique used 

before, we do not claim to have created it. It is a simple technique based on the bisection method. The bisection method is explained in section 4.2.1 and a simple explanation of the multi-dimensional multi-section (MM) method based on the bisection method is also found in section 4.2.1. Both methods are used to find the maximum value and if the entire volume had a single local maximum, both methods would yield the same result. Nevertheless, this work is an uncharted area and we do not know for sure if this is the case. To reduce the probability of finding a local maxima that is not the maximum of the entire volume, in the first level of search we look at many smaller regions. This still does not guarantee that we find the absolute maximum, but it is more reliable than the bisection method.

A careful check of the overall paper is advised to remove the typos and grammatical errors.

We are committed to always deliver the best writing we possibly can, sometimes double checking or even triple checking the paper using different tools at our disposal. Also, we are fortunate to have a native English speaker in the team.

Round 2

Reviewer 6 Report

Comments and Suggestions for Authors

The authors have fully considered my former suggestions. In this case, I'd like to accept it for publication in its current form.

Comments on the Quality of English Language

The English expression is OK for publication